

# A simple tool for refining GCM water availability projections, applied to Chinese catchments

Joe M. Osborne[1] and F. Hugo Lambert[1]

[1]College of Engineering, Mathematics and Physical Sciences, University of Exeter, Exeter, UK.

**Correspondence:** Joe M. Osborne (j.m.osborne@exeter.ac.uk)

**Abstract.** There is a growing desire for reliable 21st-century projections of water availability at the regional scale. Global climate models (GCMs) are typically used together with global hydrological models (GHMs) to generate such projections. GCMs alone are unsuitable, especially if they have biased representations of aridity. The Budyko framework describes how water availability varies as a non-linear function of aridity and is used here to constrain projections of runoff from GCMs, without the need for computationally expensive GHMs. Considering a Chinese case study, we first apply the framework to observations to show that the contribution of direct human impacts (water consumption) to the significant decline in Yellow river runoff was greater than the contribution of aridity change by a factor of approximately 2, although we are unable to rule out a significant contribution from the net effect of all other factors. We then show that the Budyko framework can be used to narrow the range of Yellow river runoff projections by 34 %, using a multi-model ensemble and the high end RCP8.5 emissions scenario. This increases confidence that the Yellow river will see an increase in runoff due to aridity change by the end of the 21st century. Yangtze river runoff projections change little, since aridity biases in GCMs are less substantial. Our approach serves as a quick and inexpensive tool to rapidly update and correct projections from GCMs alone. This could serve as a valuable resource when determining the water management policies required to alleviate water stress for future generations.

## 1 Introduction

Climate change is a global problem, but the impacts and associated vulnerability are not homogeneous. There is therefore a demand for robust projections of changes in regional climate, particularly water availability. At the largest scales, the majority of literature on projected changes in aridity suggests a global land drying tendency (Dai, 2013; Cook et al., 2014; Scheff and Frierson, 2015); a consequence of ubiquitous increases in potential evapotranspiration ($E_p$), but mixed signals in precipitation ($P$). At the river catchment scale, direct human impacts (non-climatic, human interventions directly affecting the partitioning of $P$ into runoff ($Q$) and evapotranspiration ($E$)) are already having a significant, but poorly quantified, effect on water availability (Nilsson et al., 2005; Gerten et al., 2008; Destouni et al., 2013; Haddeland et al., 2014). For the Indus river catchment Haddeland et al. (2014) showed that current direct human impacts on water availability (decreases due to water consumption for irrigation) are expected to be greater in magnitude than end-of-21st-century climatic impacts on water availability. Increasing $E$ due to irrigation is commonly observed in heavily populated catchments, especially across southern and eastern Asia (Gordon et al., 2005).





Projecting future water availability can be thought of as a supply and demand issue; the net atmospheric supply of water versus the net demand for water resulting from direct human impacts (land-use change, dam construction and reservoir operation, and surface water and groundwater consumption for irrigation). Recent studies have considered either: 1) the projected human water demand using integrated assessment models, with water supply fixed to present conditions (Hejazi et al., 2014a); 2) the projected water supply using global climate models (GCMs) to force offline global hydrological models (GHMs), with human water demand fixed to present conditions (Cook et al., 2014; Schewe et al., 2014); or 3) both projected water supply and projected human water demand (Haddeland et al., 2014; Hejazi et al., 2014b).

Using GCM output alone in hydrological projections is not considered suitable, because GCMs have coarse resolution, simplified land surface schemes and, crucially, biases in simulating hydrological cycle components. The usual approach for generating hydrological projections is to use bias corrected and downscaled GCM output to force offline GHMs (Wood et al., 2004). Using GHMs in addition to GCMs greatly increases the computational expense of a study. Here, we propose an approach for refining projections of water availability from GCM models participating in phase 5 of the Coupled Model Intercomparison Project (CMIP5). We use $Q$ as a measure of water availability. The term "refine" is used in the sense that we expect to generate projections of $Q$ on an improved physical footing, compared to using GCM output directly. The approach uses model simulated aridity and a bias correction, within the Budyko framework (Budyko, 1974). We do not consider future human water demand, only future net atmospheric supply of water, of which aridity is a key determinant.

We consider a simple water balance and assume that changes in storage are negligible:

$$P = Q + E. \tag{1}$$

GCM variables require bias correction to be of value at catchment scales (Schewe et al., 2014). Bias correcting a GCM simulated future $Q$ or $E$ is a complex process. To illustrate this point we introduce the Budyko framework (Budyko, 1974). Within this framework the partitioning of (annual to long-term mean) $P$ into $Q$ and $E$ scales as a non-linear function of aridity. Aridity, within the Budyko framework, is the dimensionless ratio of $E_p$ to $P$. The evaporative index, the dimensionless ratio of $E$ to $P$, is dependent on $E_p/P$. The relationship is described by the non-linear Budyko curve, which is constrained by the physical limits of the atmospheric demand for water ($E < E_p$; the red dashed 1:1 line in Fig. 1) and the atmospheric supply of water ($E < P$; the blue dashed horizontal line in Fig. 1).

The original deterministic and non-parametric Budyko formula was developed using data mainly from European river catchments (Budyko, 1974):

$$\frac{E}{P} = \left\{ \frac{E_p}{P} \tanh\left(\frac{P}{E_p}\right) \left[ 1 - \exp\left(-\frac{E_p}{P}\right) \right] \right\}^{1/2}. \tag{2}$$

The trajectory taken in the Budyko space due to a change in $P$, $E_p$ or $E$ is dependent on the initial values of these three fluxes (the mean state) (van der Velde et al., 2014). Therefore, an accurate representation of the observed climatology is important in any modelling study looking at hydrological projections, especially since changes in variables are typically small compared with climatology. If the present-day aridity is biased then the future-minus-present changes in runoff ($\Delta Q$) and evapotranspiration ($\Delta E$) will also be biased, even if the future-minus-present changes in precipitation ($\Delta P$) and evapotranspiration ($\Delta E_p$)



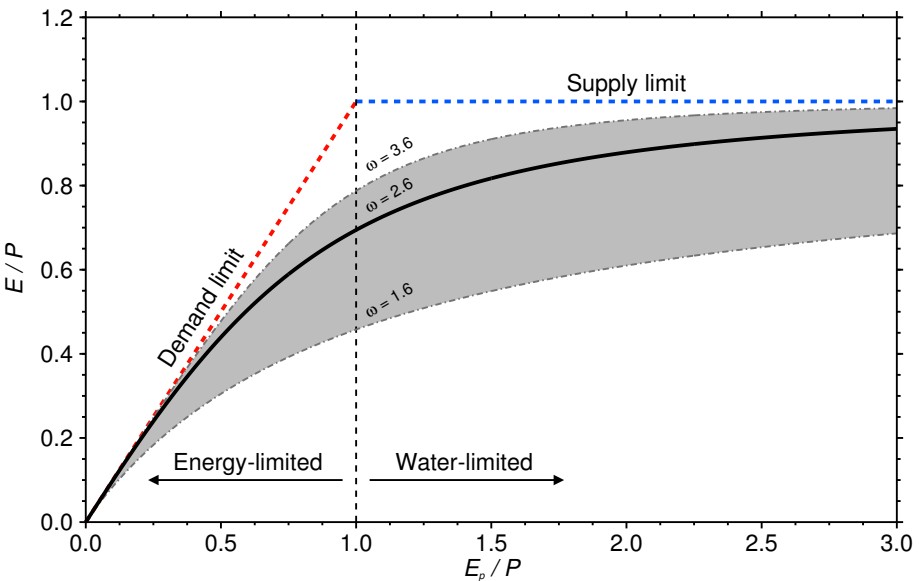

**Figure 1.** The traditional Budyko curve (solid black curve), corresponding to $\omega = 2.6$ in Eq. (3). The $\omega = 2.6 \pm 1$ curves are also shown (dot-dashed gray curves bounding the shaded region). The atmospheric supply limit ($E < P$; horizontal dashed blue line) and atmospheric demand limit ($E < E_p$; diagonal dashed red line) are shown. Energy-limited conditions are represented to the left of the vertical dashed black line ($E_p/P < 1$) and water-limited conditions are represented to the right ($E_p/P > 1$).

are correctly simulated. Supplementary Sect. S1 further demonstrates this point with an example that, although arbitrary, is illustrative of the magnitude of aridity biases in CMIP5 models.

Recent work has shown that aridity can only explain part of the differences between catchments (e.g., Zhang et al., 2001). This has led to the derivation of a number of parametric forms of the Budyko curve. One of the more popular forms is the Fu equation (Fu, 1981; Zhang et al., 2004), a one-parameter function expressed as:

$$\frac{E}{P} = 1 + \frac{E_p}{P} - \left[ 1 + \left( \frac{E_p}{P} \right)^\omega \right]^{1/\omega}, \tag{3}$$

where $\omega$ is an empirical parameter that is calibrated against local data. The traditional Budyko curve (Eq. (2)), corresponds to $\omega = 2.6$ in Eq. (3).

Here, an attempt is made to utilize biased but plentiful GCM output without the need for GHMs. We apply our approach to two major catchments in China, the Yangtze and the Yellow. The Yangtze and Yellow rivers dominate the wetter south and drier north, respectively (Fig. 2). The spatial variability in $P$ means that the north of the country, which is poleward of the East Asian monsoon rains, is more water-stressed than the south. This is exacerbated by the fact that the north has 65 % of the total arable land in China (Piao et al., 2010).





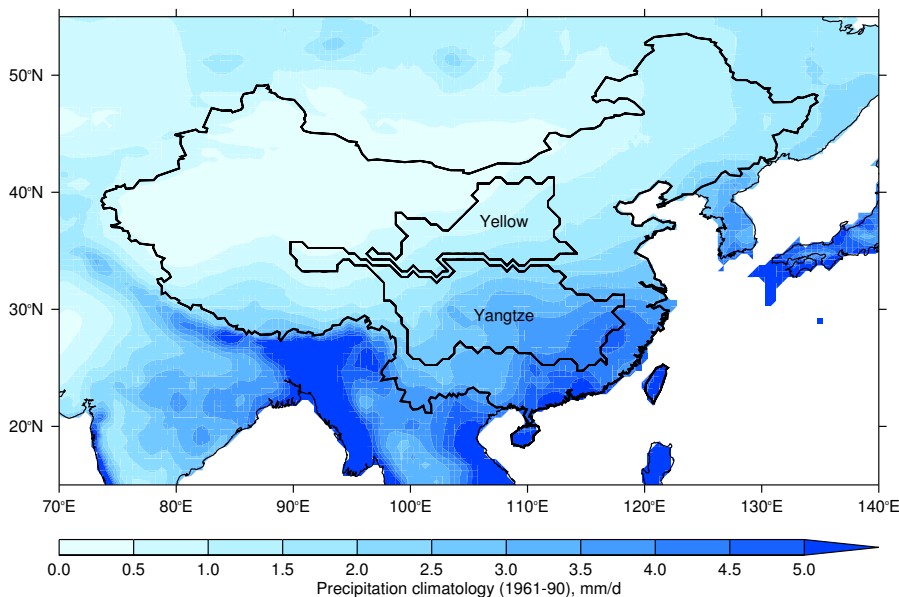

**Figure 2.** Precipitation climatology for 1961–1990 using the CRU precipitation dataset. This dataset is spatially interpolated, using available in situ observations, to give complete global land coverage. The location of the Yangtze and Yellow catchments within China is shown.

The mismatch in water supply versus water demand could be the reason behind the stark decline in Yellow river streamflow (the temporally lagged, spatial integral of upstream $Q$) seen in recent years (Yang et al., 2004). The contributions of climate change (which incorporates aridity change, but also changes in seasonality, snow dynamics and storminess (Gudmundsson et al., 2016)) and direct human impacts to this "drying up" are widely discussed in recent literature (Wang et al., 2006; Piao et al., 2010; Miao et al., 2011). However, there is little consensus on the contributions of these two components to the decrease in $Q$. Therefore, we also use the Budyko framework to quantify the contribution of aridity change alone (changes in $P$ and $E_p$ only) to the 20th century decrease in $Q$ in the Yellow catchment. We reconcile this estimate with $Q$ simulated by an offline land surface model (LSM) that does not include a representation of direct human impacts, with the exception of land-use change. This acts to test the suitability of the Budyko framework for this attribution. Further, we ask if the difference between the total change in $Q$ and the component attributed to aridity change, for the Yellow catchment, is in close agreement with a simple estimate of the change in $Q$ due to direct human impacts.

Section 2.1 details the observed and modelled data used and Sect. 2.2 describes the methodology. Results are presented in Sect. 3, first applying the Budyko framework to the 20th century observed water availability (Sect. 3.1), before extending the approach to constrain 21st-century model projections of water availability (Sect. 3.2). We finish with a discussion (Sect. 4) and conclusions (Sect. 5).





## 2 Data and methods

### 2.1 Data

We use the Dai et al. (2009) Global River Flow and Continental Discharge Dataset to calculate observed $Q$ for the Yangtze and Yellow catchments. This dataset aimed to use the farthest downstream gauging station (to maximise spatial representation) that had good temporal coverage. $Q$ is calculated by dividing river discharge at a gauging station by the upstream catchment area. In keeping with many other hydrological studies we use annual mean values throughout, but consider the water year (October-September). Data are available for October 1950 to September 2000.

To ensure an accurate comparison between observed $P$ and $Q$ we produce high-resolution catchment masks on a $0.5° \times 0.5°$ grid to match that of the $P$ dataset used (Fig. 2). We select the latest Climatic Research Unit (CRU) high-resolution $P$ dataset, CRU TS3.23 (Harris et al., 2014). The interpolated version of the dataset is used, which offers complete global terrestrial coverage. This allows for direct comparison with the spatial and temporal coverage of observed Yangtze and Yellow $Q$. By restricting our analysis of observations to 1951–2000 we find that our conclusions are not sensitive to using either the interpolated version or raw version of the precipitation dataset (see supplementary Sect. S2 and Fig. S1). We then calculate $E$ as $P - Q$ (Eq. (1)).

Likewise, we use the CRU TS3.23 $E_p$ dataset ($0.5° \times 0.5°$ resolution), which is estimated from variables such as temperature, vapour pressure, cloud cover and wind speed, using a variant of the Penman-Monteith equation. This $E_p$ estimator is computed from variables that are often poorly observed, both spatially and temporally. An energy-only $E_p$ estimator would be preferable (Sheffield et al., 2012; Milly and Dunne, 2016), but required observations are not available.

We also use $Q$ output from the Lund-Potsdam-Jena (LPJ) LSM (Sitch et al., 2003; Osborne et al., 2015). This is forced over the 1951–2000 historical period with observed CRU $P$, as well as other observed CRU climate variables (Harris et al., 2014) and changing $CO_2$ concentrations (more details are given in supplementary Sect. S3). The run considered was also driven by historical land-use changes, calculated from the History Database of the Global Environment (HYDE) (Klein Goldewijk and Verburg, 2013). A separate run excludes the HYDE dataset, so that we are able to test the sensitivity to land-use changes. Simulated $Q$ is available at a monthly frequency at $0.5° \times 0.5°$ resolution. The LPJ LSM is chosen from a multi-model ensemble that forms the TRENDY intercomparison project (Sitch et al., 2013) because it simulates a long-term mean (1951–2000) runoff coefficient ($Q/P$) that is closest to that observed for both major Chinese river catchments.

We use data from 34 GCMs participating in CMIP5 (Taylor et al., 2012). These are listed in supplementary Sect. S4. We consider data for historical (1951–2005) and two 21st-century RCP emissions scenarios (RCP4.5 and RCP8.5; 2006–2100) experiments. Only one ensemble member was used for each model and experiment (the first; r1i1p1). Simulated data are regridded to $0.5° \times 0.5°$ resolution and masked to the two Chinese river catchments. We calculate $Q$ as $P - E$ (Eq. (1)). However, data for $Q$ is also directly available for 28 of the 34 GCMs. Conclusions should not be sensitive to using either direct $Q$ output or water balance-derived $Q$ if changes in storage are negligible. Bring et al. (2015), however, showed evidence for long-term systematic changes in water storage in some CMIP5 models. It is sensible to test the sensitivity of our results to the choice of $Q$.





An energy-only $E_p$ estimator is used for CMIP5 models. $E_p$, being a hypothetical construct, is not a standard output of CMIP5 models. We follow recent work (e.g., Greve et al., 2014; Greve and Seneviratne, 2015; Milly and Dunne, 2016) and estimate $E_p$ directly from net surface radiation ($R_n$):

$$E_p = \frac{R_n}{\lambda}, \tag{4}$$

where $\lambda$ is the latent heat of vapourisation ($\lambda \approx 2.45$ MJ kg$^{-1}$). This simple energy-only $E_p$ estimator has been shown to
perform well compared to more complicated estimators, particularly under significant climate change (Sheffield et al., 2012).

## 2.2 Methods

The Budyko framework can be used to estimate the aridity change contribution to the overall change in $Q$. We have to first calibrate $\omega$ against local data for each catchment. Using the observed annual mean $P$, $E$ and $E_p$ for 1951–1960, $\omega$ is calculated as the value that minimises the mean squared errors between the observed annual mean $E/P$ ratios and those modelled using
Eq. (3), for each catchment. Following Li et al. (2013) the objective function is:

$$Obj = min \sum_i \left\{ \frac{E_i}{P_i} - \left\{ 1 + \frac{(E_p)_i}{P_i} - \left[ 1 + \left( \frac{(E_p)_i}{P_i} \right)^\omega \right]^{1/\omega} \right\} \right\}^2, \tag{5}$$

where $i$ is the year. The period 1951–1960, in this context, is considered to be representative of natural $Q$ (minimal water consumption or regulation by human activities). There will be some direct human impacts on $Q$ at this time, with a substantial Chinese land area equipped for irrigation even in the 1950s (Freydank and Siebert, 2008), although it does pre-date major dam construction; the Sanmenxia dam was the first major dam in the Yellow catchment and was completed in 1960. In calculating
the $E_p/P$ (aridity change) contribution to the change in $E/P$ (Eq. (3)) we take $\omega$ to be constant over the period 1951–2000. Our results are not qualitatively affected by the length of period chosen to represent natural $Q$ (analyses are repeated with 5, 15 and 20 year periods, all starting in 1951).

We use $\omega$ values of 1.74 and 2.29 for the Yangtze and Yellow, respectively. Combining Eq. (1) with Eq. (3) gives:

$$Q_a = -E_p + P \left\{ \left[ 1 + \left( \frac{E_p}{P} \right)^\omega \right]^{1/\omega} \right\}, \tag{6}$$

where $Q_a$ is the runoff due to aridity change (changes in $P$ and $E_p$ only) and so $\omega$ is taken to be constant. This separates aridity
change from changes in all other climatic factors besides aridity change, as well as changes in all non-climatic factors. All other climatic and non-climatic factors are integrated by $\omega$. This aridity change component is sometimes referred to as the natural $Q$ in other studies (Wang et al., 2006). However, this can be misleading since changes in $P$ and $E_p$ include both changes due to natural variability and, potentially, human-induced changes (Zhang et al., 2007; Dai, 2013).

We also estimate the runoff due to direct human impacts ($Q_h$) for the Yellow catchment only, since previous work suggests
that $Q_h$ contributes significantly to the measured runoff ($Q_m$) here (Wang et al., 2006; Miao et al., 2011). Time series of water consumption are derived to estimate $Q_h$. Water consumption is defined as the water withdrawn for human use that leaves a catchment (Xu et al., 2010). Agricultural sector irrigation accounts for a large proportion of total water consumption and, in



turn, $Q_h$. A year 2000 water consumption estimate of 0.082 mm day$^{-1}$ for the Yellow catchment (48 % of the 1951–1960 mean $Q_m$) (Xu et al., 2010) is scaled with a 1951–2000 time series of Chinese irrigated area (Freydank and Siebert, 2008). Irrigated area in China increased three-fold between 1951 and 2000 and we assume that Yellow catchment irrigated area has changed in proportion with national changes. Accurate quantification of past (and even present) water consumption is immensely difficult,

but using estimates of past irrigated area offers a means of making pseudo-quantitative statements about $Q_h$.

Defining the change in runoff due to aridity change as $\Delta Q_a$, the measured change in runoff ($\Delta Q_m$) can be approximated as the sum of $\Delta Q_a$, the change in runoff due to direct human impacts ($\Delta Q_h$) and the change in runoff due to all other climatic and non-climatic factors besides aridity change and direct human impacts ($\Delta Q_o$):

$$\Delta Q_m = \Delta Q_a + \Delta Q_h + \Delta Q_o, \tag{7}$$

with changes over the historical period (1951–2000) calculated as the linear trend. The Budyko framework can only separate

the contribution of aridity change to the measured decrease in Yellow river runoff from the contribution of all other factors besides aridity change (time-varying $\omega$), represented by the residual $\Delta Q_h + \Delta Q_o$ in Eq. (7). The parameter $\omega$ integrates all other factors, so a significant residual represents a significant net contribution from these factors. Changes in climatic factors besides aridity, such as seasonality, snow dynamics and storminess, and non-climatic factors besides direct human impacts, such as land surface characteristics and the physiological response of plants to increasing $CO_2$ ($CO_2$ fertilization, $CO_2$ stomatal closure and

water-use efficiency) could all play a role. However, previous literature suggests that $\Delta Q_h$ has been significant in the Yellow catchment. We therefore decompose the residual in Eq. (7) into a component due to direct human impacts and a component due to all other factors besides both aridity change and direct human impacts. Since water is being diverted from the river and heavily consumed, we expect $\Delta Q_h$ to be negative.

We reconcile $Q_a$ with $Q$ simulated by the LPJ LSM. Although the LSM is unable to simulate water resources with the

complexity of a GHM, it does include a representation of some of the factors integrated by $\omega$, particularly non-climatic factors such as changes in land-use and land cover, the response of stomata to rising $CO_2$ concentrations, $CO_2$ fertilization and soil moisture controls on transpiration (see supplementary Sect. S3 and Sitch et al. (2013)). Different runs test the sensitivity to land-use change. The representation of these other factors means that we do not truly compare like-for-like when reconciling $Q_a$ with $Q$ simulated by the LPJ LSM. However, we still expect aridity change to be the dominant driver of runoff in the LPJ

LSM and so define the change in runoff simulated by the LPJ LSM as $\Delta Q_{a_l}$.

Equation (6) is also used to constrain projections of $Q$ in CMIP5 models, instead substituting $P$ with a corrected $P$ ($P'$) and $E_p$ with a corrected $E_p$ ($E_p'$):

$$Q^* = -E_p' + P' \left\{ \left[ 1 + \left( \frac{E_p'}{P'} \right)^{\omega} \right]^{1/\omega} \right\}, \tag{8}$$

where $Q^*$, the Budyko corrected runoff, is calculated for the period 1951–2100. An asterisk (rather than a prime) is used to show that $Q$ has been corrected using the Budyko framework and not directly using a simple bias correction. The bias correction

technique chosen to calculate $P'$ and $E_p'$ is covered in Sect. 3.2. This is because the results of exploratory data analyses on $\Delta P$ and $\Delta E_p$, and how these relate to climatology biases across the CMIP5 models, will inform the choice of correction technique.



## 3 Results

### 3.1 20th-century historical changes

The drying of the Yellow river has been one of the most notable aspects of hydrological change in China over recent decades (Yang et al., 2004; Piao et al., 2010). There has been a significant negative linear trend in Yellow river $Q$ between 1951 and 2000 (-0.26 $\pm$ 0.06 mm day$^{-1}$ century$^{-1}$, $p < 0.05$; range is the 5–95 % range, taken as $\pm$ 1.64 standard deviations), while the decrease in $P$ over the equivalent period is not significant at the 95 % ($p = 0.05$) confidence level (-0.17 $\pm$ 0.21 mm day$^{-1}$ century$^{-1}$) (Fig. 3). The decrease in $Q$ is particularly notable since about 1970. Despite a substantial human water demand in the second half of the 20th century there has been a slight, non-significant, increase in $Q$ in the Yangtze catchment (0.04 $\pm$ 0.29 mm day$^{-1}$ century$^{-1}$) that is closely matched by a slight, non-significant, increase in $P$ (0.02 $\pm$ 0.34 mm day$^{-1}$ century$^{-1}$).

The Yangtze river shows no tendency to shift towards a distinct new area of the Budyko space between 1951 and 2000 (Fig. 4). The Yellow river, however, seems to shift towards larger $E/P$ values (smaller $Q/P$). Within the Budyko framework this could be expected under a shift towards greater aridity (larger $E_p/P$ values), or increases in $\omega$. A systematic shift towards greater aridity is not obvious in Fig. 4. There is a significant positive linear trend in Yellow river $E/P$ between 1951 and 2000 (0.22 $\pm$ 0.05 per century), but the positive trend in $E_p/P$ (0.52 $\pm$ 0.50 per century) is only significant at the 90 % ($p = 0.10$) confidence level. This suggests that all other factors ($\omega$) may also be a key driver of changes in $E/P$ over this time period in the Yellow catchment. Given this evidence and the significant negative linear trend in Yellow river $Q$, we investigate further the contributions of aridity change and all other factors to the decrease in $Q$.

$\Delta Q_a$ is noticeably different to $\Delta Q_m$ for the Yellow catchment (-0.07 $\pm$ 0.08 mm day$^{-1}$ century$^{-1}$ and -0.26 $\pm$ 0.06 mm day$^{-1}$ century$^{-1}$, respectively), with a significantly less negative trend ($\Delta Q_a - \Delta Q_m$ is equal to 0.19 $\pm$ 0.07 mm day$^{-1}$ century$^{-1}$, $p < 0.05$) (Fig. 5). We reconcile our $\Delta Q_a$ calculations with $\Delta Q_{a_l}$. The linear trends are statistically consistent (-0.07 $\pm$ 0.08 mm day$^{-1}$ century$^{-1}$ and -0.05 $\pm$ 0.06 mm day$^{-1}$ century$^{-1}$ for $\Delta Q_a$ and $\Delta Q_{a_l}$, respectively). This also holds when considering the LPJ LSM run without land-use changes, for which $\Delta Q_{a_l}$ is -0.06 $\pm$ 0.06 mm day$^{-1}$ century$^{-1}$. Our results are not sensitive to fixed or varying land-use.

If aridity change and direct human impacts have dominated the measured change in Yellow river runoff, so that the change in runoff due to all other factors is negligible, from Eq. (7) we get $\Delta Q_a \approx \Delta Q_m - \Delta Q_h$. We calculate $\Delta Q_h$ as -0.11 $\pm$ 0.01 mm day$^{-1}$ century$^{-1}$ for the Yellow river (note that the uncertainty range is artificially small due to the limited temporal resolution of the irrigated area time series of Freydank and Siebert (2008)). Therefore, $\Delta Q_m - \Delta Q_h$ (-0.15 $\pm$ 0.07 mm day$^{-1}$ century$^{-1}$) does not fully reconcile our estimates of $\Delta Q_a$ and $\Delta Q_{a_l}$ (-0.07 $\pm$ 0.08 mm day$^{-1}$ century$^{-1}$ and -0.05 $\pm$ 0.06 mm day$^{-1}$ century$^{-1}$, respectively). $\Delta Q_h$ only accounts for 59 % and 54 % of $\Delta Q_m - \Delta Q_a$ and $\Delta Q_m - \Delta Q_{a_l}$, respectively. This imbalance could suggest a significant contribution from $\Delta Q_o$, or be explained by an underestimate of the year 2000 water consumption. We calculate the year 2000 water consumption that balances $\Delta Q_a = \Delta Q_m - \Delta Q_h$ to be 0.140 mm day$^{-1}$, a 70 % increase on the estimate of Xu et al. (2010). This closely matches a year 2000 water consumption estimate by Zhu et al. (2003) of 0.137 mm day$^{-1}$. Calculating the relative contribution of aridity change to the measured decrease in Yellow river runoff as ($\Delta Q_a/\Delta Q_m$) $\times$ 100% returns a value of 27 %. Using the two estimates of year 2000 water consumption of 0.082




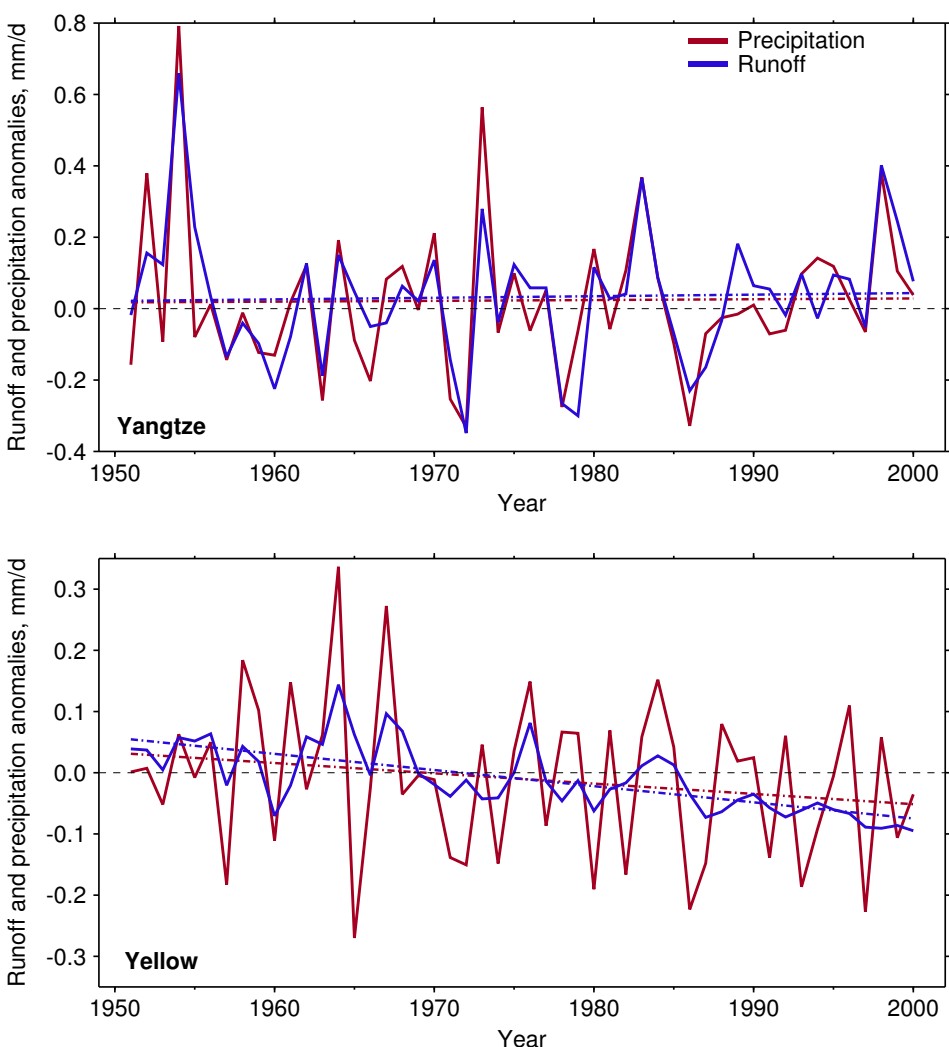

**Figure 3.** Observed runoff and precipitation anomalies for the Yangtze (top) and Yellow (bottom) river catchments for 1951–2000, relative to 1961–1990. The dot-dashed lines show linear fits to the time series.





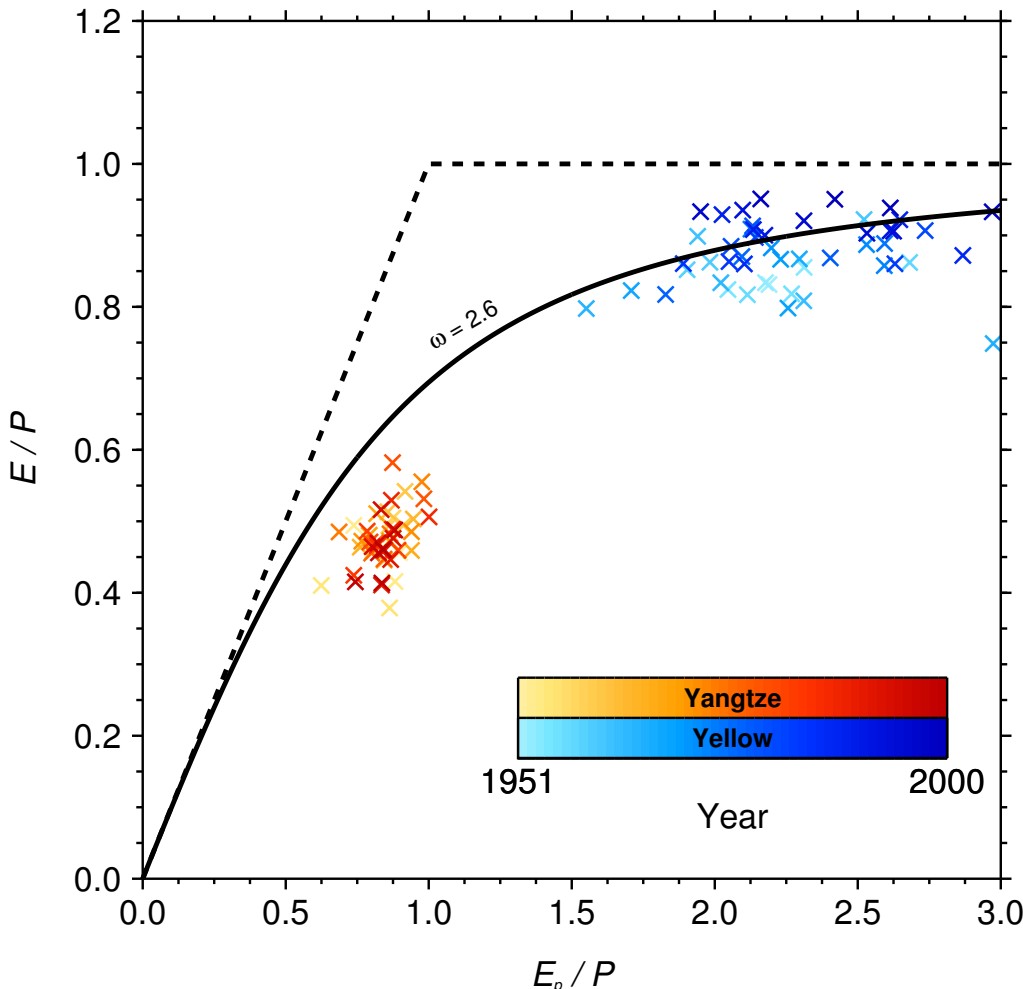

**Figure 4.** The evaporative index against aridity for the Yangtze (red) and Yellow (blue) river catchments. The symbols represent observed annual mean data for 1951–2000 with darker shades for the more recent years. The traditional Budyko curve is fitted, corresponding to $\omega = 2.6$ in Eq. (3).



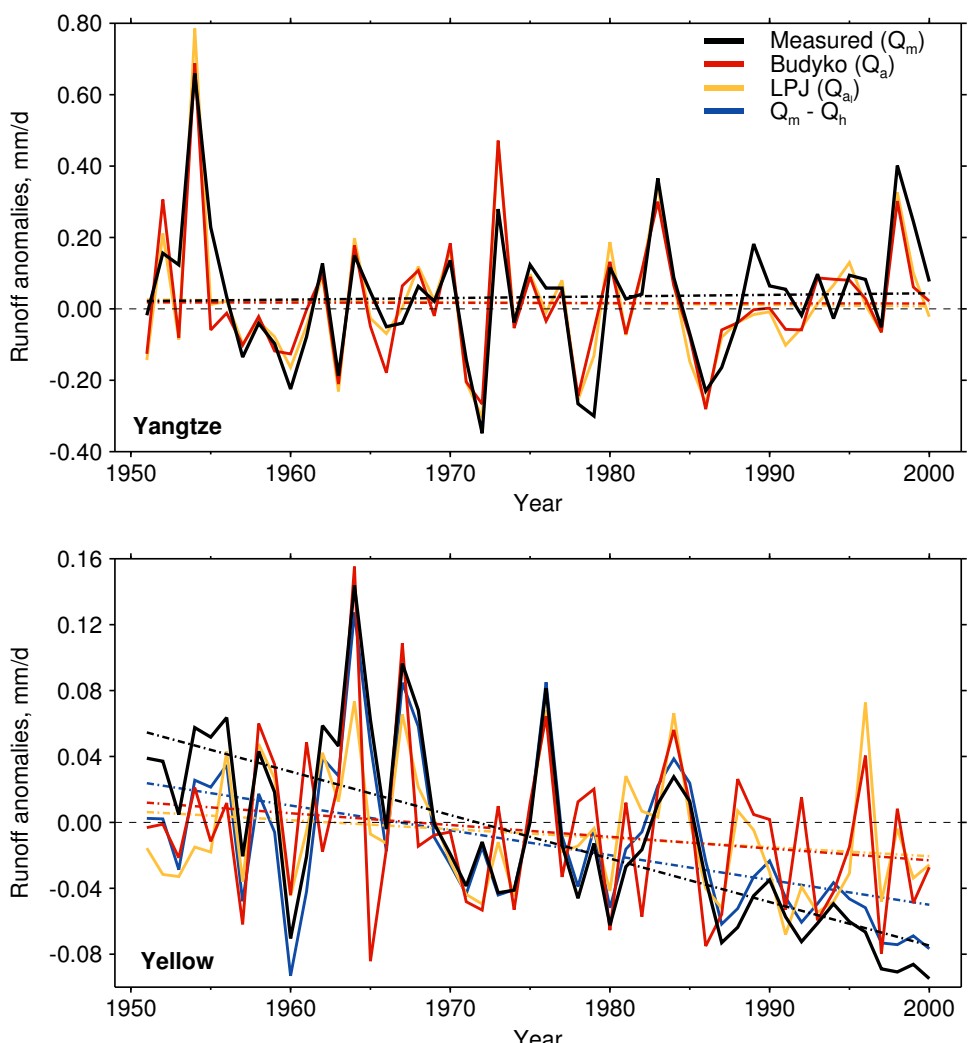

**Figure 5.** Runoff anomalies for the Yangtze (top) and Yellow (bottom) river catchments for 1951–2000, relative to 1961–1990. Shown are measured runoff ($Q_m$), runoff due to aridity change ($Q_a$), runoff simulated by the LPJ LSM ($Q_{a_l}$) and (for the Yellow river only) the difference between $Q_m$ and runoff due to direct human impacts ($Q_h$). The dashed lines show linear fits to the time series.



mm day$^{-1}$ and 0.137 mm day$^{-1}$, the relative contribution of direct human impacts to the measured decrease in Yellow river runoff $((\Delta Q_h/\Delta Q_m) \times 100\%)$ is 43 % and 71 %, respectively.

We account for between 70 % and 98 % of $\Delta Q_m$ with $\Delta Q_a + \Delta Q_h$, using the low and high water consumption estimates, respectively. Using this information with Eq. (7), we could suggest that the contribution from $\Delta Q_o$ is either negligible (using

the high water consumption estimate) or significant (using the low water consumption estimate). Instead, it shows that there is considerable uncertainty in quantifying water consumption and, in turn, the contribution of $\Delta Q_h$ to $\Delta Q_m$. Nevertheless, the close agreement of $\Delta Q_a$ and $\Delta Q_{a_l}$ suggests that direct human impacts have played a larger role than aridity change in causing the water availability crisis in the Yellow catchment. The contribution of direct human impacts would appear to be greater, by a factor of approximately 2, than the contribution of aridity change. It is worth remembering that $\Delta Q_a$ will not only reflect

natural variability, but also human-induced changes; $E_p$ has increased due to human-induced warming (Dai, 2013) and $P$ has changed due to various anthropogenic forcings (Osborne and Lambert, 2014; Burke and Stott, 2017).

## 3.2 21st-century projected changes

From the Budyko framework, changes in $Q$ are not only dependent on changes in $P$, $E_p$ and $E$, but also the initial values of these three fluxes. This means that we should view $Q$ projections cautiously if there are biases in key hydrological cycle

variables in CMIP5 models. Consistent with previous work (Chen and Frauenfeld, 2014), we find that the spatial pattern of $P$ over China is reproduced by CMIP5 models but annual mean $P$ is overestimated in most regions, compared to CRU climatology. This is evident in the multi-model mean $P$ bias (Fig. 6), with the greatest wet biases seen in the the western parts of the Yangtze and Yellow catchments (the eastern Tibetan Plateau).

As a result of these $P$ biases most CMIP5 models do not fall in the same region of the Budyko space as observations for

the Yellow catchment (Fig. 7). Although $P$ is overestimated in the Yangtze catchment for 1951–2000 (3.78 ± 0.97 mm day$^{-1}$ and 2.74 mm day$^{-1}$ for CMIP5 and observations, respectively), there is little multi-model mean bias in $E_p/P$ (0.88 ± 0.28 and 0.85 for CMIP5 and observations, respectively), implying that $E_p$ is also overestimated (3.24 ± 0.47 mm day$^{-1}$ and 2.30 mm day$^{-1}$ for CMIP5 and observations, respectively). In contrast, there is considerable multi-model mean bias in $E_p/P$ in the Yellow catchment (1.35 ± 0.52 and 2.27 for CMIP5 and observations, respectively), with models (on average) simulating a

humid rather than a semi-arid climate zone, according to a widely-used aridity classification (Middleton and Thomas, 1997). In fact, only one of 34 models considered simulates an aridity greater than 2.0 (MRI-CGCM3). This misrepresentation is a result of a significant overestimate of $P$ for 1951–2000 (2.18 ± 0.91 mm day$^{-1}$ and 1.10 mm day$^{-1}$ for CMIP5 and observations, respectively) and a less biased simulation of $E_p$ (2.75 ± 0.43 mm day$^{-1}$ and 2.45 mm day$^{-1}$ for CMIP5 and observations, respectively).

Figure 8 shows the multi-model mean $\Delta P$ and $\Delta E_p$ (using 1980–1999 and 2080–2099 as present-day and future climates, respectively) in RCP8.5. Consistent with previous literature (Chen and Frauenfeld, 2014), $P$ increases in CMIP5 projections throughout China, with significant increases across most of the Yellow catchment. $P$ also increases across the Yangtze catchment, although fewer models simulate significant increases here. As discussed by Greve and Seneviratne (2015), significant $E_p$





**Figure 6.** Absolute (top) and relative (bottom) multi-model mean precipitation climatology bias for 1961–1990. The location of the Yangtze and Yellow catchments within China is shown. Desert regions ($<200$ mm year$^{-1}$), as determined from CRU climatology (1961–1990), are masked in white.





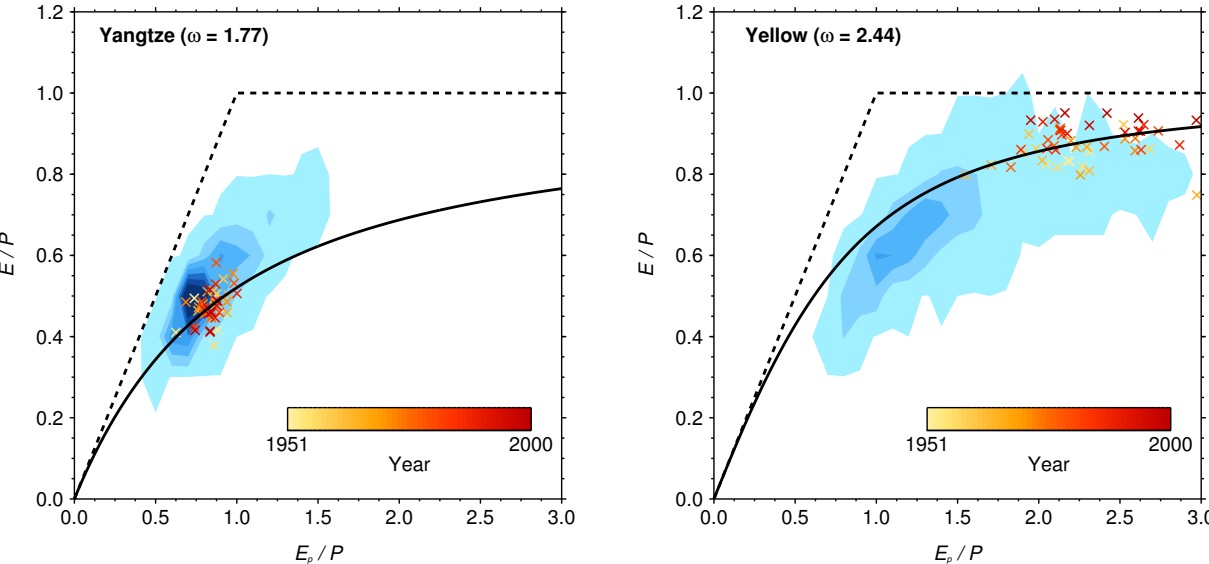

**Figure 7.** The evaporative index against aridity for the Yangtze (left) and Yellow (right) river catchments. The shaded blue regions represent the density of CMIP5 annual mean data for the 1951–2000 period, with darker shades meaning more data in a given region of the Budyko space. The symbols represent observed data with darker shades for the more recent years. $\omega$ values are calculated for the 1951–2000 period using Eq. (5).

increases are ubiquitous. CMIP5 multi-model mean $\Delta P$ is $0.36 \pm 0.56$ mm day$^{-1}$ and $0.35 \pm 0.30$ mm day$^{-1}$ for the Yangtze and Yellow, respectively. Respective values for $\Delta E_p$ are $0.33 \pm 0.23$ mm day$^{-1}$ and $0.25 \pm 0.19$ mm day$^{-1}$.

    Although we expect model simulated $\Delta Q$ and $\Delta E$ to be erroneous due to climatology biases (as highlighted in the hypothetical example in supplementary Sect. S1), we assume that $\Delta P$ and $\Delta E_p$ in the CMIP5 multi-model ensemble are not dependent

5 on the biases in climatology described above (Fig. 6). Figure 9 shows how $\Delta P$ and $\Delta E_p$ relate to the climatology of $P$ ($\overline{P}$) and $E_p$ ($\overline{E_p}$), respectively, across the 34 CMIP5 models. There are weak but significant correlations between $\Delta P$ and $\overline{P}$ in RCP8.5 for both the Yangtze and Yellow catchments, but significance is lost with the exclusion of an outlying model in each case. The weak but significant correlation between $\Delta P$ and $\overline{P}$ in RCP4.5 for the Yangtze catchment is also dependent on an outlying model. There is little evidence for significant correlations between $\Delta E_p$ and $\overline{E_p}$.

10     If there was strong evidence for relationships between $\Delta P$ and $\overline{P}$ and/or $\Delta E_p$ and $\overline{E_p}$ then a simple multiplicative correction, applied to catchment annual mean $P$ and $E_p$, would be appropriate (Hempel et al., 2013). For $P$:

$$P'_{GCM} = P_{GCM} \times \frac{\overline{P}_{CRU}}{\overline{P}_{GCM}}, \tag{9}$$





**Figure 8.** Multi-model mean future-minus-present changes (2080–2099 minus 1980–1999) in $P$ (top) and $E_p$ (bottom) in RCP8.5. Stippling indicates where less than 50 % of the CMIP5 models show significant change, as determined with a $t$-test comparing present-day and future climates. Absence of stippling indicates where more than 50 % of the models show significant change and more than 80 % of the significant models agree on the sign. Gray indicates where more than 50 % of the models show significant change but less than 80 % of the significant models agree on the sign. This method follows (Tebaldi et al., 2011). Desert regions ($<$200 mm year$^{-1}$), as determined from CRU climatology (1961–1990), are masked in white for $P$.





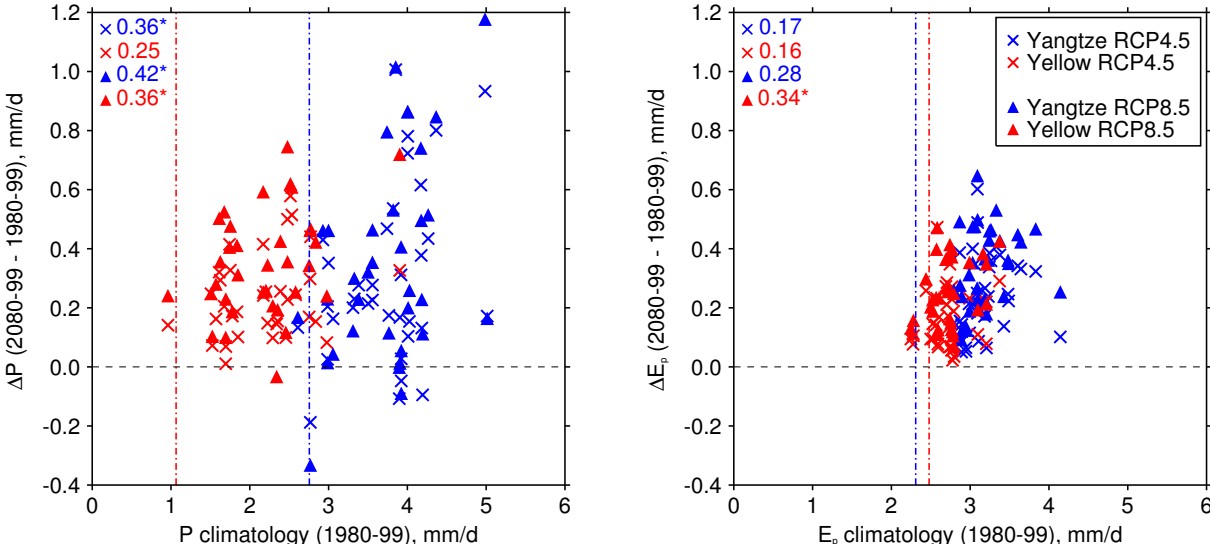

**Figure 9.** Future-minus-present changes (2080–2099 minus 1980–1999) in $P$ (left) and $E_p$ (right) against $P$ and $E_p$ climatologies (1980–1999), respectively, for 34 CMIP5 models. The dashed vertical lines show the observed climatologies for the Yellow (red) and Yangtze (blue) catchments. The values in the top left of each panel refer to correlation coefficients for the catchment and RCP emissions scenario listed in the legend. Those with asterisks are significant at the 95 % ($p = 0.05$) confidence level.

where the subscript GCM is an individual model from the CMIP5 ensemble, the subscript CRU is the observed data and $P'$ is the corrected $P$. The period 1980–1999 is used to calculate the climatologies. Using a multiplicative correction factor preserves the relative rather than absolute trends in model simulated $P$ and $E_p$.

CMIP5 $P$ and $E_p$ biases for the Yangtze and Yellow catchments are, on average, positive and substantial (Fig. 9). As such, the correction factor in Eq. (9) is, on average, less than unity. A multiplicative correction factor would therefore narrow the ranges of $\Delta P$ and $\Delta E_p$ across the CMIP5 ensemble. The assumption that we can use absolute $\Delta P$ and $\Delta E_p$ from CMIP5 models seems valid in the absence of strong evidence for relationships between $\Delta P$ and $\overline{P}$ and/or $\Delta E_p$ and $\overline{E_p}$. We instead use a simple additive correction (Hempel et al., 2013). Temporally constant offsets (the absolute differences between observed and simulated climatologies) are added to model simulated $P$ and $E_p$. For $P$:

$$P'_{GCM} = P_{GCM} + (\overline{P}_{CRU} - \overline{P}_{GCM}). \tag{10}$$

We adjust $P$ and $E_p$ in the 34 CMIP5 models for 1951–2100 to eliminate the biases in simulating the observed CRU climatologies, while retaining absolute $\Delta P$ and $\Delta E_p$. Positivity constraints on $P$ and $E_p$ can render additive corrections inappropriate, but this is not a problem at the spatial (catchment) and temporal (annual) resolutions considered here.

$Q$ (as simulated by the CMIP5 models and calculated using $P - E$) differs considerably from $Q^*$ (Fig. 10) as calculated with Eq. (8), particularly for the Yellow catchment. The Budyko corrected future-minus-present change in runoff ($\Delta Q^*$; recall that



**Table 1.** CMIP5 model simulated ($\Delta Q$) and CMIP5 Budyko corrected ($\Delta Q^*$) future-minus-present runoff changes (mm day$^{-1}$) for 2080–2099, relative to 1980–1999. The multi-model mean and 5–95 % ranges across the individual models are listed (based on a Gaussian assumption). For comparison, values for a subset of 28 (from 34) CMIP5 models for which $Q$ is directly simulated are also shown. CMIP5 model directly simulated future-minus-present runoff changes ($\Delta Q_{direct}$) are used to verify the suitability of calculating $\Delta Q$ as $\Delta P - \Delta E$ (water balance-derived).

|  |  | RCP4.5 | RCP8.5 |
|---|---|---|---|
| Yangtze (all): | $\Delta Q$ | $0.12 \pm 0.32$ | $0.14 \pm 0.40$ |
|  | $\Delta Q^*$ | $0.16 \pm 0.33$ | $0.18 \pm 0.39$ |
| Yangtze (subset): | $\Delta Q$ | $0.08 \pm 0.26$ | $0.09 \pm 0.35$ |
|  | $\Delta Q^*$ | $0.12 \pm 0.26$ | $0.13 \pm 0.32$ |
|  | $\Delta Q_{direct}$ | $0.08 \pm 0.25$ | $0.10 \pm 0.33$ |
| Yellow (all): | $\Delta Q$ | $0.07 \pm 0.11$ | $0.09 \pm 0.14$ |
|  | $\Delta Q^*$ | $0.06 \pm 0.07$ | $0.09 \pm 0.09$ |
| Yellow (subset): | $\Delta Q$ | $0.06 \pm 0.11$ | $0.09 \pm 0.15$ |
|  | $\Delta Q^*$ | $0.06 \pm 0.07$ | $0.09 \pm 0.10$ |
|  | $\Delta Q_{direct}$ | $0.06 \pm 0.11$ | $0.09 \pm 0.16$ |

the future-minus-present change is the mean of 2080–2099 minus the mean of 1980–1999) is similar to $\Delta Q$ for the Yangtze catchment in both RCP4.5 and RCP8.5 across the CMIP5 ensemble (Table 1). In the Yellow catchment (RCP8.5) the multi-model mean $\Delta Q^*$ matches that of the multi-model mean $\Delta Q$ (both 0.09 mm day$^{-1}$). The 5–95 % range, however, is reduced by 34 % ($\pm$ 0.14 mm day$^{-1}$ to $\pm$ 0.09 mm day$^{-1}$). Similar results are found with RCP4.5, with little change in the multi-model mean from 0.07 mm day$^{-1}$ to 0.06 mm day$^{-1}$ but a decrease of 35 % in the 5–95 % range from $\pm$ 0.11 mm day$^{-1}$ to $\pm$ 0.07 mm day$^{-1}$ for $\Delta Q$ and $\Delta Q^*$, respectively. These findings are not sensitive to using directly simulated runoff instead (Fig. 10 and Table 1). The small differences between $Q$ and $Q^*$, and $\Delta Q$ and $\Delta Q^*$, for the Yangtze catchment are expected given that CMIP5 models broadly fall in the correct region of the Budyko space (Fig. 7). For the Yellow catchment, the uncertainties in runoff projections have been reduced considerably. The CMIP5 multi-model mean $\Delta Q^*$ in RCP8.5 is significantly different from zero at the 90 % confidence level. Such a level of confidence is not achieved for $\Delta Q$.

## 4 Discussion

Before using the Budyko framework in tandem with CMIP5 output, we considered whether it could be used to quantify the contribution of aridity change to the measured decrease in Yellow river runoff between 1951 and 2000. Encouragingly, for both the Yangtze and Yellow catchments, the $Q$ trend due to aridity change was found to be near-identical to that simulated using





**Figure 10.** CMIP5 model simulated ($Q$; orange) and CMIP5 Budyko corrected ($Q^*$; blue) runoff anomalies for 1951–2100, relative to 1980–1999, for the Yangtze (top) and Yellow (bottom) river catchments in the historical and RCP8.5 experiments. Shown are the 5-year running multi-model mean (thick line) and 5–95 % ranges (shading) across the CMIP5 ensemble. The box plots (mean, ± one standard deviation ranges, 5–95 % ranges, and minimum to maximum ranges) are given for 2080–2099 (Table 1). Also shown, for comparison, are box plots for a subset of 28 (from 34) CMIP5 models for which $Q$ is directly simulated (not limited to being calculated as $P - E$). The unfilled box plot shows CMIP5 model directly simulated runoff for 2080–2099.





the LPJ LSM (which is forced by observed $P$ and $E_p$). This suggests that the Budyko framework is suitable for determining the relative contribution of aridity change to the measured decrease in Yellow river runoff, calculated as 27 %. Therefore, the relative contribution of all other factors besides aridity to the measured decrease in Yellow river runoff is expected to equal 73 %.

Deriving time series of water consumption using low and high year 2000 water consumption estimates, the component due to direct human impacts is calculated as 43 % and 71 %, respectively. Therefore, we can account for nearly all of the measured decrease in Yellow river runoff (98 %) using aridity change and the high consumption estimate alone, but stress that such estimates are highly uncertain. We are not able to dismiss a significant contribution from the net effect of all other factors (besides aridity and direct human impacts), which ranges from 2 % to 30 %. Given that the estimate of the contribution of
aridity change appears to be the most robust result, we can instead state that the majority of the measured decrease in Yellow river runoff appears to be due to direct human impacts and all other factors. Also, despite the uncertain water consumption estimates, the contribution from direct human impacts is approximately 2 times greater than the contribution from aridity change. Other studies have estimated the climate change (all non-human) and human components. Miao et al. (2011) attribute 55 % of the reduction in Yellow river water discharge to humans, with Wang et al. (2006) giving a value of 49 %, compared to
our range of 43 % to 71 %. Note that these studies use different methods and periods to estimate the contributions of the two components, but focus on the second half of the 20th century. Our estimate of the component due to direct human impacts is consistent with these previous estimates, although we add detail by finding that this contribution is markedly greater than the contribution from aridity change alone.

Although estimates of water consumption are highly uncertain there are also uncertainties in our estimate of the aridity
change contribution to $Q$ change. This estimate, as well as runoff simulated by the LPJ LSM, rely on an uncertain observed $E_p$ dataset (see Sect. 2.1). An energy-only $E_P$ estimator is expected to be more appropriate (Milly and Dunne, 2016), but is not available because of insufficient observed data. Meanwhile, the observed $P$ dataset is likely to contain biases and inhomogeneities (Osborne and Lambert, 2014). Many grid boxes in China are poorly gauged (some not at all) in the period investigated (see supplementary Fig. S1), especially in the mountainous Tibetan Plateau region, where $P$ is scarce but highly
variable (Adam et al., 2006). These are largely insurmountable obstacles facing all hydroclimatological studies.

Within the Budyko framework all climatic and non-climatic factors besides aridity are integrated by the $\omega$ parameter. In Eq. (7) we separate this "residual" into a component due to direct human impacts and a component due to all other climatic and non-climatic factors besides aridity change and direct human impacts. The low water consumption estimate means that we are not able to dismiss a significant contribution from the net effect of all other factors. Support for a negligible contribution from
all other factors comes from the strength of agreement between $Q_a$ and $Q_{a_l}$. This is because the LPJ LSM includes a realistic representation of vegetation, which is known to be a useful integrated indicator of these other factors that are integrated by $\omega$ (Li et al., 2013) (see supplementary Sect. S3). Further, supplementary Fig. S3 shows that CMIP5 models simulate no obvious changes in $\omega$ over the second half of the 20th century.

In estimating direct human impacts from just water consumption there remains the possibility that other direct human impacts
could account for a significant contribution to the decrease in Yellow river runoff. We present evidence that the contribution





from land-use change is negligible. On the other hand, catchment runoff can abruptly decrease during the filling of large reservoirs following dam construction, causing anomalously low annual runoff. Following filling, runoff should return to pre-dam levels and such projects are only thought to affect seasonal water storage and not introduce trends in long-term runoff. Rather, dam and reservoir construction facilitates access to water resources and leads to more water withdrawal and consumption. The

influence of dams and reservoirs are likely accounted for in the water consumption estimates (Biemans et al., 2011).

The agreement between the Budyko framework and the LPJ LSM for the observed period also increases our confidence in using the Budyko framework for projections. The CMIP5 Budyko corrected projected changes in runoff rely on the assumption that 21st-century changes in $P$ and $E_p$ are not dependent on existing climatology biases in CMIP5 models. Across the CMIP5 multi-model ensemble we did not find compelling evidence for relationships, supporting this assumption (Fig. 9). This is

broadly consistent with expectations, given recent research showing that the "wet gets wetter, dry gets drier" paradigm (Held and Soden, 2006) does not hold over global land surfaces (Greve et al., 2014; Greve and Seneviratne, 2015). However, the mean state can undoubtedly have some influence on the simulated changes in $P$ and $E_p$ due to land-atmosphere feedbacks (Berg et al., 2016). We note that when correcting $E_p$ (Eq. (10); with $P$ replaced with $E_p$) we calculate the correction offset as the observed climatology (Penman-Monteith estimator) minus the model simulated climatology (energy-only estimator).

Using these different estimators will likely introduce some error in the calculation.

In calculating $Q^*$ (Eq. (8)) $\omega$ values are calculated for the 1951–2000 period, using Eq. (5), then taken to be constant for the period 1951–2100. While the relationships of variations in $Q$ with variations in such catchment-specific parameters are understood (Roderick and Farquhar, 2011; Gudmundsson et al., 2016), the full complexity of the influence of changes in catchment properties on these parameters is not. However, Li et al. (2013) showed that, for large catchments, the long-term

averaged annual vegetation coverage explains as much as 63 % of the variance in the catchment-specific $\omega$. With 21st-century increases in total vegetation coverage projected (Schneck et al., 2015), we expect this parameter will increase in magnitude. This is found to be the case in the CMIP5 multi-model ensemble and these increases in $\omega$ need to be included when verifying the Budyko framework on the CMIP5 models themselves (see supplementary Sect. S3 and Figs. S2-4). The influence of changes in $\omega$ on projected changes in $Q$ is small compared to the influence of correcting $E_p$ and $P$ (see supplementary Sect. S3 and

Fig. S5). Demonstrating this, the CMIP5 multi-model mean $\Delta Q^*$ for the Yellow catchment in RCP8.5 with constant $\omega$ (0.09 ± 0.09 mm day$^{-1}$) is not significantly different to the CMIP5 multi-model mean $\Delta Q^*$ for the Yellow catchment in RCP8.5 with time-varying $\omega$ (0.07 ± 0.08 mm day$^{-1}$). Therefore, our conclusions are not sensitive to the choice of $\omega$ (constant or time-varying).

We show that aridity change (changes in $P$ and $E_p$ only) is of greatest importance in shaping projected changes in runoff

in CMIP5 models and all other factors ($\omega$) play a secondary role. We expect our CMIP5 Budyko corrected $Q$ projections to be substantially more reliable than the original CMIP5 model simulated $Q$ projections. In the case of the Yellow catchment, the future-minus-present (2080–2099 minus 1980–1999) change in $Q$ 5–95 % range is reduced by 34 % and 35 % in RCP8.5 and RCP4.5, respectively. Importantly, constraining $Q$ projections using the Budyko framework increases confidence that the Yellow catchment will see increases in $Q$ by the end of the 21st century – the best guess (CMIP5 multi-model mean) change

of 0.09 mm day$^{-1}$ is significantly different from zero at the 90 % confidence level. Greater confidence in the range of Yellow





catchment water availability projections could be of great value to policy-makers. More generally, the Budyko framework serves as an inexpensive tool to rapidly update projections from biased GCM simulations without the need for offline GHMs. However, further research is needed. Specifically, we believe that an ensemble of GHMs, driven by at least one set of bias corrected and downscaled GCM projections, should be used as a means of verification.

**5  Conclusions**

We have demonstrated how the Budyko framework can be used to place water availability projections from readily-available GCM output onto a more physical basis by correcting for biases in aridity, using the example of the Yangtze and Yellow river catchments in China. The approach is inexpensive, does not need the use of offline GHMs, and could be used to provide rapid updates on water availability projections for new GCM scenarios. Wherever GCMs simulate significant biases in representing

observed aridity, we expect to generate significantly altered projections. In the Yellow catchment, considerable negative biases in simulated aridity lead to a substantial narrowing of the range of future GCM projections. In catchments where GCMs simulate positive biases, we would expect to see broadening of the range of GCM projections. Meanwhile, in the Yangtze catchment, simulated aridity biases are small, meaning that projections are little changed by our approach.

We stress again that these refined water availability projections account for aridity change only. In the hypothetical case

where future aridity change is known, the projected $Q$ will not be realised due to the effect of all other factors, especially highly uncertain future changes in direct human impacts (these are not represented in CMIP5 models). Current human impacts on $Q$ are possibly greater than end-of-21st-century aridity change impacts on $Q$ in the Yellow catchment (Haddeland et al., 2014). Therefore, the current water shortages are not likely to be alleviated without improved agricultural practices and water management. Importantly though, reducing the range of water availability projections gives planners an improved idea of what

needs to be done to reduce water stress in the Yellow catchment for future generations. Moreover, our conclusions underline the need for imminent action and highlight the fact that increases in $Q$ due to aridity change will not offer much relief in the absence of serious and concerted action to minimize direct human impacts.

Chinese authorities have recently attempted to alleviate the drying in the north of China, by diverting water there from the wetter south (the South-to-North Water Diversion Project). It remains to be seen whether this will reduce the imbalance in

atmospheric water supply and human water demand across China and whether it could even place additional water stress on the more resilient south (Barnett et al., 2015). Generating refined water availability projections in these two key river catchments should underpin decisions made on future engineering projects.

*Data availability.*  The CMIP5 data can be accessed via the web portal https://esgf-node.llnl.gov/projects/esgf-llnl/. Sources for the observed datasets are: CRU TS3.23 $P$ and $E_p$ (Harris et al., 2014) http://www.cru.uea.ac.uk/data/; Global River Flow and Continental Discharge

Dataset $Q$ (Dai et al., 2009) http://www.cgd.ucar.edu/cas/catalog/surface/dai-runoff/. Data from the LPJ LSM experiments are available from the authors upon request.



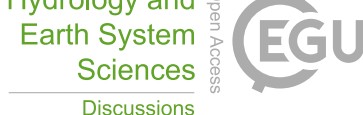

*Competing interests.* The authors declare that they have no conflict of interest.

*Acknowledgements.* This work was supported by the Natural Environment Research Council grant NE/M006123/1 and the UK-China Research & Innovation Partnership Fund through the Met Office Climate Science for Service Partnership (CSSP) China as part of the Newton Fund. We acknowledge the World Climate Research Programme's Working Group on Coupled Modeling, which is responsible for CMIP, and we thank the climate modeling groups for producing and making available their model output. For CMIP the U.S. Department of Energy's Program for Climate Model Diagnosis and Intercomparison provides coordinating support and led development of software infrastructure in partnership with the Global Organization for Earth System Science Portals.



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
