# Peer review of "A simple tool for refining GCM water availability projections, applied to Chinese catchments"

_Hydrology and Earth System Sciences, 2018_

## Referee Comment (RC1) · Anonymous Referee #1 · 28 May 2018

The authors propose an approach to refine future projections of Q by using simulated aridity and biased corrections via the Budyko framework. They aim to generate Q projections that are much better than Q data derived from direct GCM projections. Using the Budyko framework to refine Q data and correct biases seems like an interesting idea. However, I think that several big questions remain regarding the methodology and that need to be answered before the article can be published. The authors have made an effort in trying to explain extensively the calculations performed, however, this has been done in a way that is hard for the reader to understand. Also, I could not understand the relation of the methods with the "bias-correction" that the authors want to apply.

[Figure]

I have several questions:

1. The term "biased correction" is not defined thoroughly in the study. I read the article but I could not relate the biased calculations with the methodology described by the authors. It was just hard to follow. A proper explanation of what specific "biases" are the author trying to correct is missing. For instance, are the authors trying to correct the CMIP5 Q data with a Budyko-type equation (Eq. 2). If this is it, I do not understand why are there using so many ways of calculating Q and related changes (climate, human, GCM-LSM, Qa, Qh, below combined) in the context of this study. Please explain this clearly.

Also regarding what I just mentioned, if the authors are trying to perform this bias correction of CMIP5 data, I must say that I have a feeling that CMIP5 data for Q is already biased-corrected. For bias corrected I mean that it is made at least consistent in Budyko space ($0 < ET/P < 1$ and $PET/P > 0$ and $PET/P > ET/P$) for most basins in the world. Strangely, the direct CMIP5 ET data does not comply with this (all over Budyko space), and hence I assume there has been some type of "bias correction" in this sense for CMIP5 Q data. The authors should check Q and ET data from CMIP5 data for their two basins and if so, please update.

2. The authors describe the methods partly in the introduction, partly under "Data" and partly under "methods". This was confusing, and hard to follow in general. I would describe the methodology in chronological order and only under "Methods". In this way, you would also clear much needed space to expand the literature review which is now limited. Please mention the several studies that use the Budyko framework to understand water changes in Chinese basins. I mention here a few. The authors say that there is "little consensus on the contributions of these two components to the decrease in Q". I would say that there is plenty, mainly afforestation and/or flow regulation. And what is the Qh have to do with the bias correction. Again, please expand on this.

[Figure]

These Budyko-China basins studies could be included.

• Fang, J., Chen, A., Peng, C., Zhao, S. and Ci, L.: Changes in Forest Biomass Carbon Storage in China Between 1949 and 1998, Science, 292(5525), 2320–2322, doi:10.1126/science.1058629, 2001.

• Huang, M., Zhang, L. and Gallichand, J.: Runoff responses to afforestation in a watershed of the Loess Plateau, China, Hydrol. Process., 17(13), 2599–2609, doi:10.1002/hyp.1281, 2003.

• Jaramillo, F. and Destouni, G.: Local flow regulation and irrigation raise global human water consumption and footprint, Science, 350(6265), 1248–1251, doi:10.1126/science.aad1010, 2015.

• Liu, M., Tian, H., Chen, G., Ren, W., Zhang, C. and Liu, J.: Effects of Land-Use and Land-Cover Change on Evapotranspiration and Water Yield in China During 1900-20001, JAWRA Journal of the American Water Resources Association, 44(5), 1193–1207, doi:10.1111/j.1752-1688.2008.00243.x, 2008.

• Qiu, G. Y., Yin, J., Tian, F. and Geng, S.: Effects of the "Conversion of Cropland to Forest and Grassland Program" on the Water Budget of the Jinghe River Catchment in China, J. Environ. Qual., 40(6), 1745–1755, doi:10.2134/jeq2010.0263, 2011.

• Xu, X., Yang, D., Yang, H. and Lei, H.: Attribution analysis based on the Budyko hypothesis for detecting the dominant cause of runoff decline in Haihe basin, Journal of Hydrology, 510, 530–540, doi:10.1016/j.jhydrol.2013.12.052, 2014.

• Zhang, X., Zhang, L., Zhao, J., Rustomji, P. and Hairsine, P.: Responses of streamflow to changes in climate and land use/cover in the Loess Plateau, China, Water Resour. Res., 44(7), W00A07, doi:10.1029/2007WR006711, 2008.

3. I tried to understand the methods:

a. You calculate historical Q from GRDC data (1951-2000)

b. You calculate E as P-Q from a) (1951-2000)

c. You calculate Ep from Penman-Monteith (1951-2000)

d. You calculate Q from LPJ-LSM (1951-2000). Here I could not understand what is this estimate trying to represent? Is it land-use driven Q, climatic Q, combined Q, or what? It was hard to follow, the explanation of the multiple runs.

e. You calculate Q from the CMIP5 data (2006-2100) as P-E. Here see my comment 1, specially regarding the statement of line 31 Page 5, "Conclusions should. . ."

f. You calculate Ep from the CMIP5 data (2006-2100)

g. Calibrate Eq. 2 to obtain w. How did you do that? I would do it as:

*Wang, D. and Hejazi, M.: Quantifying the relative contribution of the climate and direct human impacts on mean annual streamflow in the contiguous United States, Water Resources Research, 47(10), n/a–n/a, doi:10.1029/2010WR010283, 2011.

h. You calculate Qa

i. You calculate Qh, I could not understand why nor how. What does Qh have to do with the biased correction?

j. Now you calculate changes in all Q components.

k. Then you compare Qa with Q from LPJ LSM. Again, it is hard to know what this comparison should result in, since it is not clear what Q from LPJ LSM really represent.

l. I got lost in what Q* psychically means. If this is the main purpose of the study, then I cannot understand why the authors go through a to k.

The authors need to mention what calculations are related with the bias-correction, and which ones are related with the aim of calculating the human component of Q (Qh). So, is this the chronological order of calculations? If not, please modify. Also, where is the correction bias coming into these methods, I could not see it, until maybe

the calculation of Q*. Please specify.

---

## Referee Comment (RC2) · Anonymous Referee #2 · 29 May 2018

The manuscript entitled "A simple tool for refining GCM water availability projections, applied to Chinese catchments" by Joe Osborne et al., presents a simple tool to post-process CMIP5 model output in order to constrain projections of water availability. The tool is based on the Budyko framework and uses bias-corrected estimates of modeled precipitation and potential evaporation to obtain catchment runoff for two large basins in China: the Yangtze and the Yellow River. The approach was shown to substantially reduce uncertainties in the runoff projections, especially for the Yellow river basin. It was further shown that observed runoff changes within the second half of the 20th century are primarily caused by human interventions.

The manuscript is generally in a good shape, well structured and well written. The overall presentation of the results is good with concise and high-quality figures. The methodological approach seems to be technically sound, but (due to its complexity) needs to be explained better. Maybe some sort of conceptual figure or flow chart would help! Further, the obtained results depend on many assumptions which potentially not permit a robust interpretation of the results. The authors already discuss some limitations of their approach, but it is my assessment that this discussion needs to be extended before final publication. Also, some important references are missing to better outline some issues and limitations of their approach.

Major comments:

1) I would be very careful with separating the measured change in runoff into the individual components as done in eq. 7. If you assume such a linear relationship, you also assume the individual components to be independent, which they are clearly not! Especially the separation into $Q_h$ and $Q_o$ is potentially dangerous. Please also be aware that in the context of the Budyko framework, aridity is solely defined through the notional, dimensionless ratio $E_p/P$, which has no direct physical meaning. Everything else besides mean annual $E_p/P$ is actually integrated into $w$. Also, $w$ and $E_p/P$ are not necessarily independent (please see Padron et al, 2017). It would be nice if you could try to determine if there are dependencies between $Q_a$, $Q_h$, and $Q_o$. Is it possible to plot these against each other? In case there are large dependencies and interrelationships the obtained results might be less meaningful.

2) In the main text, the authors assume $w$ (omega) to be constant. In the Supplementary they further present results obtained for a (time-)varying $w$. I actually leave this to the authors, but I would almost prefer to present the time-varying more prominently. A certain variation around the original Budyko curve and thus a variation of $w$ is actually inherent to the Budyko framework. This was already stated by Budyko himself. Hence, the Budyko framework is not necessarily deterministic. There is a growing body of literature interpreting the Budyko framework in a probabilistic sense (e.g. Greve et al.,

2015, Singh and Kumar, 2015, Gudmundsson et al., 2016, etc.), thus accounting for the spread in w and taking into account the complex interplay of all other factors (besides the aridity index). By using the time-varying approach you basically account for these variations, which in my assessment is more realistic.

3) In this context, please be aware and discuss that you are considering temporal variations here. Most of the referenced Budyko-based studies actually consider spatial variations. It is important to note that these are not necessarily tradable (Berghuijs and Woods, 2017).

4) You use the Budyko-based equation 8 to compute Q* for each year. One of the main assumptions of the Budyko hypothesis is stationarity, i.e. storage changes are negligible. This might, however, not be the case here due to interannual variations in storage. Have you maybe tested to use longer time periods to smooth out long-term variations in water storage? How would your results look like when computing decadal Q*?

Minor comments:

p. 1, l. 3: I would not use "describe" here, it might be better to use something like "represent". The Budyko framework does not mechanistically describe the relationship between the aridity and the evaporative index.

p. 1, l. 16-19: There is some recent literature which challenges the referenced papers and the ubiquitous increase in aridity in a warmer world (Roderick et al., 2015, Scheff et al., 2017, Greve et al., 2017, Scheff, 2018). Might be worth to mention that this is an ongoing debate.

p. 2, l. 1-3: It might be better to rephrase this a bit. Depending on the context, the terms "water supply" and "water demand" are interpreted differently. Water supply is not necessarily just atmospheric water supply. Water supply can also be runoff. Some people also consider groundwater or water from other, unconventional sources

(water transfer, desalination, etc.) as water "supply". Also, water stored in reservoirs is some sort of water supply. Water demand is often used in terms of human water consumption, including domestic and industrial water use as well as water used for irrigation.

p. 3 l. 7: For w=2.6, please cite Yang et al, 2008.

p. 4, Fig. 2: The basin outlines look a bit strange. I know that this is based on grid cells, but why don't you just outline the grid cells instead of interpolating the basin borders. This would also help to avoid the gap between both basins

p. 4, l. 3: Among seasonality, snow dynamics and storminess it is also many other factors that are related to a hydroclimatological change. Maybe just add an "etc."

p. 5, l. 24-26: Is this shown by Sitch et al., 2013. In this case, it might be better to add the reference at the end of the sentence. Or is this based on a separate analysis? In this case, you might add something like "not shown"?

p. 6, eq. 4: Milly actually proposes an ad-hoc multiplier of 0.8. Have you checked if you can maybe reduce some of the biases in Ep by adjusting the multiplier?

p. 8, l. 30: See my first major comment. This might also be due to dependencies between each factor.

p. 18, Fig. 10: Is the time series for Q* computed annually? And is the 5yr running mean subsequently computed from the annual time series? Or is Q* computed from the 5yr running mean time series of P and Ep?

p. 19, l. 30-32: Please note here that Li et al. (2018) is potentially only valid at large catchment scales. At other scales, this might not be the case. You might also note here or somewhere else that the interpretation of w is still largely and inconclusive and sometimes also contradictory (Padron et al., 2017).

References:
Padrón, Ryan S., Lukas Gudmundsson, Peter Greve, and Sonia I. Seneviratne. 2017. "Large-Scale Controls of the Surface Water Balance Over Land: Insights From a Systematic Review and Meta-Analysis". Water Resources Research 53 (11): 9659–78. https://doi.org/10.1002/2017WR021215.

Greve, P., L. Gudmundsson, B. Orlowsky, and S. I. Seneviratne. 2015. "Introducing a Probabilistic Budyko Framework". Geophysical Research Letters 42 (7): 2015GL063449. https://doi.org/10.1002/2015GL063449.

Gudmundsson, L., P. Greve, and S. I. Seneviratne. 2016. "The Sensitivity of Water Availability to Changes in the Aridity Index and Other Factors - a Probabilistic Analysis in the Budyko-Space". Geophysical Research Letters, Januar, 2016GL069763. https://doi.org/10.1002/2016GL069763.

Singh, Riddhi, and Rohini Kumar. o.ÂăJ. "Vulnerability of Water Availability in India Due to Climate Change: A Bottom-up Probabilistic Budyko Analysis". Geophysical Research Letters 42 (22): 9799–9807. https://doi.org/10.1002/2015GL066363.

Berghuijs, Wouter R., and Ross A. Woods. 2016. "Correspondence: Space-Time Asymmetry Undermines Water Yield Assessment". Nature Communications 7 (Mai): 11603. https://doi.org/10.1038/ncomms11603.

Roderick, Michael L., Peter Greve, and Graham D. Farquhar. 2015. "On the Assessment of Aridity with Changes in Atmospheric CO2". Water Resources Research 51 (7): 5450–5463. https://doi.org/10.1002/2015WR017031.

Scheff, Jacob, Richard Seager, Haibo Liu, and Sloan Coats. 2017. "Are Glacials Dry? Consequences for Paleoclimatology and for Greenhouse Warming". Journal of Climate 30 (17): 6593–6609. https://doi.org/10.1175/JCLI-D-16-0854.1.

Greve, Peter, Michael L. Roderick, and Sonia I. Seneviratne. 2017. "Simulated Changes in Aridity from the Last Glacial Maximum to 4xCO 2". Environmental Research Letters 12 (11): 114021. https://doi.org/10.1088/1748-9326/aa89a3.

Scheff, Jacob. 2018. "Drought Indices, Drought Impacts, CO2, and Warming: A Historical and Geologic Perspective". Current Climate Change Reports 4 (2): 202–9. https://doi.org/10.1007/s40641-018-0094-1.

Yang, Hanbo, Dawen Yang, Zhidong Lei, und Fubao Sun. 2008. "New Analytical Derivation of the Mean Annual Water-Energy Balance Equation". Water Resources Research 44 (3) https://doi.org/10.1029/2007WR006135.
* * *

---

## Author Comment (AC1) · 20 Jul 2018

**Thank you for reviewing our paper and providing suggestions that have improved it. Our responses to your comments are in bold font.**

The manuscript is generally in a good shape, well structured and well written. The overall presentation of the results is good with concise and high-quality figures. The methodological approach seems to be technically sound, but (due to its complexity) needs to be explained better. Maybe some sort of conceptual figure or flow chart would help! Further, the obtained results depend on many assumptions which potentially not permit a robust interpretation of the results. The authors already discuss some

limitations of their approach, but it is my assessment that this discussion needs to be extended before final publication. Also, some important references are missing to better outline some issues and limitations of their approach.

**The second referee has also commented that the methodological approach could be explained better. We will therefore add a flow chart (Figure 1) to the end of the introduction section in the revised manuscript to serve as an overview for the subsequent breakdown of the paper.**

**The key message from this is that the Budyko curve, or rather the updated Fu version (equation 3), can be used to both attribute past changes and help refine future changes. These two strands share common ideology but are attempting to tackle different problems and therefore use different equations. The flow chart more clearly emphasises this and even lists the equations that are used in each application. We will also divide the Data (section 2.1) and Methods (section 2.2) subsections into further subsections to more clearly differentiate the two applications.**

**MANUSCRIPT TO BE AMENDED.**

**Major comments:**

1) I would be very careful with separating the measured change in runoff into the individual components as done in eq. 7. If you assume such a linear relationship, you also assume the individual components to be independent, which they are clearly not! Especially the separation into Qh and Qo is potentially dangerous. Please also be aware that in the context of the Budyko framework, aridity is solely defined through the notional, dimensionless ratio Ep/P, which has no direct physical meaning. Everything else besides mean annual Ep/P is actually integrated into w. Also, w and Ep/P are not necessarily independent (please see Padron et al, 2017). It would be nice if you could try to determine if there are dependencies between Qa, Qh, and Qo. Is it possible to plot these against each other? In case there are large dependencies and interrelationships

the obtained results might be less meaningful.

**This is a very good point and something that we have explored further. Unfortunately it is difficult to test for dependencies between $Q_h$ and $Q_o$ due to the limited temporal resolution (decadal mean values) of the irrigated area time series of Freydank and Siebert (2008) that is used to calculate $Q_h$ (page 8, line 25). We can however look at the relationship between $Q_a$ and changes dues to all other factors besides aridity change ($\omega$), represented by the residual $\triangle Q_h + \triangle Q_o$ in equation 7. The interannual correlation is -0.35 (Figure 2).**

**However, this correlation becomes positive (0.66) when considering 5 year means (Figure 3). Therefore, we do not believe there is strong and consistent evidence for such a relationship here. We do feel it is important to point this out as a potential limitation of decomposing changes in runoff into these separate components. We will therefore add the following to the discussion:**

**"It is also important to note some potential limitations of using Eq. (7) to separate the measured decrease in Yellow river runoff into various components. This approach assumes a linear relationship and therefore that the individual components are independent. Padron et al. (2017) showed that cross correlations exist between many of the factors proposed to influence runoff through $\omega$. Testing for dependencies between $\triangle Q_h$ and other components is unfortunately limited by the poor temporal resolution of the irrigated area time series of Freydank and Siebert (2008). Although we find that interannual variations in $Q_a$ and the residual $Q_h + Q_o$ are correlated (-0.35), this correlation is weak and not robust to using multi-year means. Further, our approach considers long-term trends/changes in runoff, which means that any dependencies at interannual timescales should not influence conclusions."**

**MANUSCRIPT TO BE AMENDED.**

2) In the main text, the authors assume w (omega) to be constant. In the Supplementary they further present results obtained for a (time-)varying w. I actually leave this to the authors, but I would almost prefer to present the time-varying more prominently. A certain variation around the original Budyko curve and thus a variation of w is actually inherent to the Budyko framework. This was already stated by Budyko himself. Hence, the Budyko framework is not necessarily deterministic. There is a growing body of literature interpreting the Budyko framework in a probabilistic sense (e.g. Greve et al., 2015, Singh and Kumar, 2015, Gudmundsson et al., 2016, etc.), thus accounting for the spread in w and taking into account the complex interplay of all other factors (besides the aridity index). By using the time-varying approach you basically account for these variations, which in my assessment is more realistic.

**We were equally unsure during the writing of this paper whether to present the time-varying analysis in the manuscript or whether to leave it as supplementary material. Although, as you say, using the time-varying approach allows a more complete assessment of changes within the Budyko framework our main aim here is to illustrate the large improvements that can be made by considering CMIP5 output without the large aridity biases that are currently present. The choice of $\omega$ (constant or time-varying) actually plays a small role in shaping changes compared to correcting for aridity biases ($E_p/P$). Also, many of the other factors that determine $\omega$ are not well represented in CMIP5 models so that a full consideration of $\omega$ is difficult. Ultimately, we understand the argument for both choices and would even welcome an extra opinion on this subject!**

3) In this context, please be aware and discuss that you are considering temporal variations here. Most of the referenced Budyko-based studies actually consider spatial variations. It is important to note that these are not necessarily tradable (Berghuijs and Woods, 2017).

**This is an important point to make. We would like to point out that the year-to-year variability calculated through the Budyko framework and presented in Figs. 5 and 10 should not be taken at face value, since changes in storage could**

dominate at these timescales. The Budyko framework is more appropriately applied to study long-term mean changes, which is exactly what we do with the CMIP5 projected changes. Although we present year-to-year variability in Fig. 10, the conclusions are drawn from the differences between two 20-year means at the end of the 20th and 21st centuries. Likewise, we look at trends in 20th century analysis. It is worth pointing out however that the conclusions on historical changes regarding the contributions of aridity change and direct human impacts are qualitatively the same if considering difference between 10 (or 20) year means at the start and beginning of the 1951–2000 period. We will point this out in the text by adding the following to the end of the discussion:

"Most applications of the Budyko framework consider spatial rather than temporal variations. Berghuijs and Woods (2017) demonstrate that spatial and temporal variations are not necessarily tradable. We stress that the Budyko framework is not employed here to robustly determine interannual variability in water availability, but is instead used to understand long-term trends (Sect. 3.2) or the difference between 20 year means at the end of the 20th and 21st centuries (Sect. 3.2)."

And the following to the text underneath equation 7:

"...with changes over the historical period (1951–2000) calculated as the linear trend. We note that our conclusions are not affected by using the difference between either 10 or 20 year means at the beginning and end of the historical period."

**MANUSCRIPT TO BE AMENDED.**

4) You use the Budyko-based equation 8 to compute Q* for each year. One of the main assumptions of the Budyko hypothesis is stationarity, i.e. storage changes are negligible. This might, however, not be the case here due to interannual variations in storage. Have you maybe tested to use longer time periods to smooth out long-term

variations in water storage? How would your results look like when computing decadal Q*?

**Because we have no information on changes in storage when we calculate $Q$ as $P - E$ (and assume changes in storage are equal to zero) our results do not change through taking decadal means (Figure 4).**

**We actually check the potential impact of storage changes by considering values for a subset of 28 (from 34) CMIP5 models for which $Q$ is directly simulated. This information is shown in Fig. 10 and Table 1 and does not affect conclusions, especially for the Yellow catchment, as pointed out in the results.**

**Minor comments:**

**These are all useful comments and we will act on all of them. We discuss below the comments that require a little more attention to show how we will act on them:**

p. 2, l. 1-3: It might be better to rephrase this a bit. Depending on the context, the terms "water supply" and "water demand" are interpreted differently. Water supply is not necessarily just atmospheric water supply. Water supply can also be runoff. Some people also consider groundwater or water from other, unconventional sources (water transfer, desalination, etc.) as water "supply". Also, water stored in reservoirs is some sort of water supply. Water demand is often used in terms of human water consumption, including domestic and industrial water use as well as water used for irrigation.

**We agree that this was poor wording. Rather than considering the entire problem in this context we rather meant that most studies working on water availability projections consider very specific components of supply and demand. We have rewritten the opening sentence as this:**

**"Literature on future water availability projections has typically been framed**

**around the net atmospheric supply of water versus the net demand for water resulting from direct human impacts (land-use change, dam construction and reservoir operation, and surface water and groundwater consumption for irrigation)...""**

**MANUSCRIPT TO BE AMENDED.**

p. 6, eq. 4: Milly actually proposes an ad-hoc multiplier of 0.8. Have you checked if you can maybe reduce some of the biases in Ep by adjusting the multiplier?

**This does affect $E_p$ biases. The multi-model mean bias for $E_p$ for the Yellow catchment changes from +0.30 mm/day to -0.25 mm/day relative to the observed climatology. But since the aridity biases are largely a result of precipitation biases, this makes little difference to the overall results ('Figure' 5; compare to Table 1 of the manuscript).**

p. 18, Fig. 10: Is the time series for Q* computed annually? And is the 5yr running mean subsequently computed from the annual time series? Or is Q* computed from the 5yr running mean time series of P and Ep?

**Yes, it is computed annually and the running mean is computed from the annual time series.**
* * *
[Figure]

**Introducing ideas
Section 1**

The partitioning of precipitation
(Eq. (1)) and the non-parametric
and one-parameter versions of
the Budyko curve (Eqs. (2) and (3),
respectively).

**Applying the
Budyko framework**

**20th century historical changes
Sections 2.1a, 2.2a and 3.1**

1) Use Eq. (5) to calibrate $\omega$, using observed $P$, $E_p$ and $E$ ($E$ is calculated using observed $P$ and $Q$ (Eq. (1)).

2) Calculate $Q_a$ using Eq. (6).

3) Estimate $Q_h$ using time series of water consumption derived from time series of Chinese irrigated area.

4) Separate the measured runoff changes into $Q_a$, $Q_h$ and a residual term using Eq. (7).

5) Use $Q$ as simulated by a LSM, to test the calculation of $Q_a$.

**21st century projected changes
Sections 2.1b, 2.2b and 3.2**

1) Estimate $E_p$ in CMIP5 models from net surface radiation (Eq. (4)).

2) Use Eq. (5) to calibrate $\omega$, using observed $P$, $E_p$ and $E$ ($E$ is calculated using observed $P$ and $Q$ (Eq. (1)).

3) Bias correct $P$ and $E_p$ using Eq. (10).

4) Use bias corrected $P$ and $E_p$, together with the calculated $\omega$ values to calculate $Q^*$, a Budyko corrected runoff. This uses Eq. (8).

**Fig. 1.** Schematic of how the Budyko framework is used to improve our understanding of 20th-century historical changes and 21st-century projected changes.

**Fig. 2.** Interannual correlation.

**Fig. 3.** 5 year mean correlation.

[Figure]

**Fig. 4.** Using decadal means.

|  |  | RCP4.5 | RCP8.5 |
|---|---|---|---|
| Yangtze (all): | $\Delta Q$ | 0.12 +/- 0.32 | 0.14 +/- 0.40 |
|  | $\Delta Q^*$ | 0.18 +/- 0.34 | 0.20 +/- 0.41 |
| Yangtze (subset): | $\Delta Q$ | 0.08 +/- 0.26 | 0.09 +/- 0.35 |
|  | $\Delta Q^*$ | 0.13 +/- 0.27 | 0.15 +/- 0.34 |
|  | $\Delta Q_{direct}$ | 0.08 +/- 0.25 | 0.10 +/- 0.33 |
| Yellow (all): | $\Delta Q$ | 0.07 +/- 0.11 | 0.09 +/- 0.14 |
|  | $\Delta Q^*$ | 0.07 +/- 0.07 | 0.10 +/- 0.10 |
| Yellow (subset): | $\Delta Q$ | 0.06 +/- 0.11 | 0.09 +/- 0.15 |
|  | $\Delta Q^*$ | 0.06 +/- 0.08 | 0.10 +/- 0.10 |
|  | $\Delta Q_{direct}$ | 0.06 +/- 0.11 | 0.09 +/- 0.16 |

**Fig. 5.** The equivalent to Table 1 in the manuscript using the Milly and Dunne (2016) multiplier.

[Figure]

---

## Author Comment (AC2) · 20 Jul 2018

**Thank you for taking the time to review our paper and helping to significantly improve it, particularly with regards the way the methodology is presented. Our responses to your comments are in bold font.**

1. The term "biased correction" is not defined thoroughly in the study. I read the article but I could not relate the biased calculations with the methodology described by the authors. It was just hard to follow. A proper explanation of what specific "biases" are the author trying to correct is missing. For instance, are the authors trying to correct the CMIP5 Q data with a Budyko-type equation (Eq. 2). If this is it, I do not understand why

are there using so many ways of calculating Q and related changes (climate, human, GCM-LSM, Qa, Qh, below combined) in the context of this study. Please explain this clearly.

**The only "conventional" bias correction performed in this study is in section 3.2, using equation 10. This is used to correct $P$ and $E_p$ in the CMIP5 models so that they are in the correct part of the Budyko space, prior to accounting for changes. These corrected variables (which become $P'$ and $E_p'$) are then used to calculate $Q^*$, the Budyko corrected runoff, which is defined on page 7, line 30. We think some of the misunderstanding of this and the other methodology that you list in your other comments comes down to conflating methods, data and equations that are used for the historical 20th century analysis with those that are used for the projected CMIP5 analysis. The second referee has also commented that the methodological approach could be explained better. We will therefore add a flow chart (Figure 1) to the end of the introduction section in the revised manuscript to serve as an overview for the subsequent breakdown of the paper.**

**The key message from this is that the Budyko curve, or rather the updated Fu version (equation 3), can be used to both attribute past changes and help refine future changes. These two strands share common ideology but are attempting to tackle different problems and therefore use different equations. The flow chart more clearly emphasises this and even lists the equations that are used in each application. We will also divide the Data (section 2.1) and Methods (section 2.2) subsections into further subsections to more clearly differentiate the two applications.**

**MANUSCRIPT TO BE AMENDED.**

Also regarding what I just mentioned, if the authors are trying to perform this bias correction of CMIP5 data, I must say that I have a feeling that CMIP5 data for Q is already biased-corrected. For bias corrected I mean that it is made at least consistent

in Budyko space (0<ET/P<1 and PET/P>0 and PET/P>ET/P) for most basins in the world. Strangely, the direct CMIP5 ET data does not comply with this (all over Budyko space), and hence I assume there has been some type of "bias correction" in this sense for CMIP5 Q data. The authors should check Q and ET data from CMIP5 data for their two basins and if so, please update.

**The models will be coded in such away to avoid values that are not physically consistent, but we are almost certain that $Q$ output from CMIP5 models is not biased corrected. It should just be a direct output. We check $Q$ as directly available for 28 models with water balance-derived $Q$ and do not see any sensitivity to this, stating "These findings are not sensitive to using directly simulated runoff instead (Fig. 10 and Table 1)." Further, we do not perform direct bias correction on $Q$, only $P$ and $E_p$. The "correction" of $Q$ is within the Budyko framework which also ensures that any values are physically consistent.**

2. The authors describe the methods partly in the introduction, partly under "Data" and partly under "methods". This was confusing, and hard to follow in general. I would describe the methodology in chronological order and only under "Methods". In this way, you would also clear much needed space to expand the literature review which is now limited. Please mention the several studies that use the Budyko framework to understand water changes in Chinese basins. I mention here a few. The authors say that there is "little consensus on the contributions of these two components to the decrease in Q". I would say that there is plenty, mainly afforestation and/or flow regulation. And what is the Qh have to do with the bias correction. Again, please expand on this.

**We think a lot of the methodology section will become more clear though incorporating the changes listed above in response to your comment 1, particularly with the inclusion of the flow chart. However, we do feel that the introduction as it is contains only information that is crucial to introduce the overarching ideas and, namely, the Budyko framework. While used in the methodology, the water**

balance (equation 1) is introduced here because it is critical in discussing the partitioning of $P$ into $Q$ and $E$ and how the Budyko formula helps to understand this. While we refer to equations 2-3 in the methodology to derive further equations, they are not directly used in any calculations in the form presented in the introduction. The data section only contains information on the data used, with equation 4 simply stating that we calculate $E_p$ in the CMIP5 models directly from net surface radiation. Also the methods section is currently in chronological order in the sense that one needs to determine $\omega$ (equation 5), before calculating $Q_a$ (equation 6), before $Q_a$ can be used in the decomposition of $Q_m$ in equation 7 (with $Q_h$ also detailed in the section above equation 7). On top of the improvements to the structure of the methodology suggested above we will add some specific additional information to make things easier to follow, such as the following sentence to the paragraph underneath equation 8:

"In Eq. (8) we use $\omega$ values calculated using observed data and Eq. (5) for the 1951–2000 period (1.77 and 2.44 for the Yangtze and Yellow, respectively)."

MANUSCRIPT TO BE AMENDED.

Of the Budyko-China basins studies that you list, only one is related to the Budyko framework. However, we agree that we should mention this study and some other high profile paper that we found. To the page 3, line 10 paragraph we will add:

"There is a wealth of literature that uses the Budyko framework to understand water changes in other Chinese basins (Yang et al., 2007; Xu et al., 2014; Liang et al., 2015)."

References:

Liang, W., Bai, D., Wang, F., Fu, B., Yan, J., Wang, S., Yang, Y., Long, D., and Feng, M.: Quantifying the impacts of climate change and ecological restoration on streamflow changes based on a Budyko hydrological model in China's Loess Plateau, Water Resour. Res., 51, 6500–6519, https://doi.org/10.1002/2014WR016589, 2015.

Xu, X., Yang, D., Yang, H., and Lei, H.: Attribution analysis based on the Budyko hypothesis for detecting the dominant cause of runoff decline in Haihe basin, J. Hydrol., 510, 530–540, https://doi.org/10.1016/j.jhydrol.2013.12.052, 2014.

Yang, D., Sun, F., Liu, Z., Cong, Z., Ni, G., and Lei, Z.: Analyzing spatial and temporal variability of annual water-energy balance in nonhumid regions of China using the Budyko hypothesis, Water Resour. Res., 43, W04 426, https://doi.org/10.1029/2006WR005224, 2007.

MANUSCRIPT TO BE AMENDED.

With regards the "little consensus on the contributions of these two components to the decrease in Q", we think you are correct. Most papers suggest that human impacts have had a significant influence on runoff. We will change this sentence and add the useful references you provided:

"Most studies suggest a significant contribution of direct human impacts, including afforestation and land-use change (Huang et al., 2003; Liu et al., 2008; Zhang et al., 2008; Qiu et al., 2011), although methods and attributed contributions vary."

References:

Huang, M., Zhang, L., and Gallichand, J.: Runoff responses to afforestation in a watershed of the Loess Plateau, China, Hydrol. Processes, 17, 2599–2609, https://doi.org/10.1002/hyp.1281, 2003.

Liu, M., Tian, H., Chen, G., Ren, W., Zhang, C., and Liu, J.: Effects of land use and land cover change on evapotranspiration and water yield in China during the 20th century, J. Am. Water Resour. Assoc., 44, 1193–1207,

https://doi.org/10.1111/j.1752-1688.2008.00243.x, 2008.

Qiu, G. Y., Yin, J., Tian, F., and Geng, S.: Effects of the "Conversion of Cropland to Forest and Grassland Program" on the water budget of the Jinghe River catchment in China, J. Environ. Qual., 40, 1745–1755, https://doi.org/10.2134/jeq2010.0263, 2011.

Zhang, X., Zhang, L., Zhao, J., Rustomji, P., and Hairsine, P.: Responses of streamflow to changes in climate and land use/cover in the Loess Plateau, China, Water Resour. Res., 44, W00A07, https://doi.org/10.1029/2007WR006711, 2008.

MANUSCRIPT TO BE AMENDED.

3. I tried to understand the methods:

a. You calculate historical Q from GRDC data (1951-2000) b. You calculate E as P-Q from a) (1951-2000) c. You calculate Ep from Penman-Monteith (1951-2000) d. You calculate Q from LPJ-LSM (1951-2000). Here I could not understand what is this estimate trying to represent? Is it land-use driven Q, climatic Q, combined Q, or what? It was hard to follow, the explanation of the multiple runs. e. You calculate Q from the CMIP5 data (2006-2100) as P-E. Here see my comment 1, specially regarding the statement of line 31 Page 5, "Conclusions should. . ." f. You calculate Ep from the CMIP5 data (2006-2100) g. Calibrate Eq. 2 to obtain w. How did you do that? I would do it as: *Wang, D. and Hejazi, M.: Quantifying the relative contribution of the climate and direct human impacts on mean annual streamflow in the contiguous United States, Water Resources Research, 47(10), n/a–n/a, doi:10.1029/2010WR010283, 2011. h. You calculate Qa i. You calculate Qh, I could not understand why nor how. What does Qh have to do with the biased correction? j. Now you calculate changes in all Q components. k. Then you compare Qa with Q from LPJ LSM. Again, it is hard to know what this comparison should result in, since it is not clear what Q from LPJ LSM really represent. l. I got lost in what Q* psychically means. If this is the main purpose of the study, then I cannot understand why the authors go through a to k.

**a. We do not use GRDC data. The first line of section 2.1 states "We use the Dai et al. (2009) Global River Flow and Continental Discharge Dataset..." b-c. Yes, for the historical 20th century analysis, which will be more clearly separated in the revised manuscript. d. This is stated in the page 7, line 20 paragraph. e-f. Yes, although we actually use 1951–2100 (page 7, line 30 paragraph). g. This is explained in the page 6, line 10 paragraph and we actually use equation 5. h-i. Yes. The calculation of $Q_h$ is explained at the bottom of page 6/top of page 7. It has nothing to do with the bias correction, which concerns the projected CMIP5 analysis. This will be more clear when we produce the flow chart and clean up the methodology section as described above. j-k. Yes. The LJP LSM runoff should be a proxy for $Q_a$ calculated via the Budyko framework as detailed in the text (page 7, line 20 and supplementary Sect. S3). l. Again, this is concerning the projected 21st century CMIP5 analysis, which will become much clearer in the revised manuscript.**

The authors need to mention what calculations are related with the bias-correction, and which ones are related with the aim of calculating the human component of Q (Qh). So, is this the chronological order of calculations? If not, please modify. Also, where is the correction bias coming into these methods, I could not see it, until maybe the calculation of Q*. Please specify.

**The revised layout of the methodology will make this much clearer. We have included the equations that are used in each calculation and in each application in Figure 1.**
* * *
[Figure]

[Figure]

**Introducing ideas
Section 1**

The partitioning of precipitation
(Eq. (1)) and the non-parametric
and one-parameter versions of
the Budyko curve (Eqs. (2) and (3),
respectively).

**Applying the
Budyko framework**

**20th century historical changes
Sections 2.1a, 2.2a and 3.1**

1) Use Eq. (5) to calibrate ω, using observed $P$, $E_p$ and $E$ ($E$ is calculated using observed $P$ and $Q$ (Eq. (1)).

2) Calculate $Q_a$ using Eq. (6).

3) Estimate $Q_h$ using time series of water consumption derived from time series of Chinese irrigated area.

4) Separate the measured runoff changes into $Q_a$, $Q_h$ and a residual term using Eq. (7).

5) Use $Q$ as simulated by a LSM, to test the calculation of $Q_a$.

**21st century projected changes
Sections 2.1b, 2.2b and 3.2**

1) Estimate $E_p$ in CMIP5 models from net surface radiation (Eq. (4)) .

2) Use Eq. (5) to calibrate ω, using observed $P$, $E_p$ and $E$ ($E$ is calculated using observed $P$ and $Q$ (Eq. (1)).

3) Bias correct $P$ and $E_p$ using Eq. (10).

4) Use bias corrected $P$ and $E_p$, together with the calculated ω values to calculate $Q^*$, a Budyko corrected runoff. This uses Eq. (8).

**Fig. 1.** Schematic of how the Budyko framework is used to improve our understanding of 20th-century historical changes and 21st-century projected changes.

---

## Author Response (AR2)

**Author's response to referee comments**

Response to anonymous referee #1

Thank you for taking the time to review our paper and helping to significantly improve it, particularly with regards the way the methodology is presented. Our responses to your comments are in bold font. Where we reference pages, lines, figure and tables, this corresponds to the revised maunscipt and approximately to the tracked changes manuscript.

1. The term "biased correction" is not defined thoroughly in the study. I read the article but I could not relate the biased calculations with the methodology described by the authors. It was just hard to follow. A proper explanation of what specific "biases" are the author trying to correct is missing. For instance, are the authors trying to correct the CMIP5 Q data with a Budyko-type equation (Eq. 2). If this is it, I do not understand why are there using so many ways of calculating Q and related changes (climate, human, GCM-LSM, Qa, Qh, below combined) in the context of this study. Please explain this clearly.

**The only "conventional" bias correction performed in this study is in section 3.2, using equation 10. This is used to correct $P$ and $E_p$ in the CMIP5 models so that they are in the correct part of the Budyko space, prior to accounting for changes in these variables. These corrected variables (which become $P'$ and $E_p'$) are then used to calculate $Q^*$, the Budyko corrected runoff, which is defined on page 9, line 5. We think some of the misunderstanding of this, and the other methodology that you list in your following comments, comes down to conflating methods, data and equations that are used for the historical 20th century analysis with those that are used for the projected CMIP5 analysis. The other referee has also commented that the methodological approach could be explained better. We have therefore added a flow chart (Fig. 1; Fig.3 in the manuscript) to the end of the introduction section in the revised manuscript to serve as an overview for the subsequent breakdown of the paper.**

**The key message from this is that the Budyko curve, or rather the updated Fu version (equation 3), can be used to both attribute past changes and help refine future changes. These two strands share common ideology but are attempting to**

**Introducing ideas**
**Section 1**

The partitioning of precipitation (Eq. (1)) and the non-parametric and one-parameter versions of the Budyko curve (Eqs. (2) and (3), respectively).

[Figure]

Applying the
Budyko framework

**20th century historical changes**
**Sections 2.1.1, 2.2.1 and 3.1**

1) Use Eq. (5) to calibrate ω, using observed $P$, $E_p$ and $E$ ($E$ is calculated using observed $P$ and $Q$ (Eq. (1)).

2) Calculate $Q_a$ using Eq. (6).

3) Estimate $Q_h$ using time series of water consumption derived from time series of Chinese irrigated area.

4) Separate the measured runoff changes into $Q_a$, $Q_h$ and a residual term using Eq. (7).

5) Use $Q$ as simulated by a LSM, to test the calculation of $Q_a$.

**21st century projected changes**
**Sections 2.1.2, 2.2.2 and 3.2**

1) Estimate $E_p$ in CMIP5 models from net surface radiation (Eq. (4)) .

2) Use Eq. (5) to calibrate ω, using observed $P$, $E_p$ and $E$ ($E$ is calculated using observed $P$ and $Q$ (Eq. (1)).

3) Bias correct $P$ and $E_p$ using Eq. (10).

4) Use bias corrected $P$ and $E_p$, together with the calculated ω values to calculate $Q^*$, a Budyko corrected runoff. This uses Eq. (8).

**Agreement of 20th century changes with existing literature will validate the use of the Budyko framework in refining 21st-century projections**

Figure 1: Schematic of how the Budyko framework is used to improve our understanding of 20th-century historical changes and 21st-century projected changes.

tackle different problems and therefore use different equations. The flow chart more clearly emphasises this and even lists the equations that are used in each application. We have also divided the Data (section 2.1) and Methods (section 2.2) subsections into further subsections to more clearly differentiate the two applications.

MANUSCRIPT AMENDED.

Also regarding what I just mentioned, if the authors are trying to perform this bias correction of CMIP5 data, I must say that I have a feeling that CMIP5 data for Q is already biased-corrected. For bias corrected I mean that it is made at least consistent in Budyko space (0<ET/P<1 and PET/P>0 and PET/P>ET/P) for most basins in the world. Strangely, the direct CMIP5 ET data does not comply with this (all over Budyko space), and hence I assume there has been some type of "bias correction" in this sense for CMIP5 Q data. The authors should check Q and ET data from CMIP5 data for their two basins and if so, please update.

**The models are coded to avoid values that are not physically consistent, but we are almost certain that $Q$ output from CMIP5 models is not biased corrected. It is a direct output. We check $Q$ as directly available for 28 models with water balance-derived $Q$ (calculated with $P - E$) and do not see any sensitivity to this, stating "These findings are not sensitive to using directly simulated runoff instead (Fig. 11 and Table 1)." Further, we do not perform direct bias correction on $Q$, only $P$ and $E_p$. The "correction" of $Q$ is within the Budyko framework and we do not use CMIP5 $Q$ in this calculation. The Budyko framework ensures that calculated $Q$ values are physically consistent.**

2. The authors describe the methods partly in the introduction, partly under "Data" and partly under "methods". This was confusing, and hard to follow in general. I would describe the methodology in chronological order and only under "Methods". In this way, you would also clear much needed space to expand the literature review which is now limited. Please mention the several studies that use the Budyko framework to understand water changes in Chinese basins. I mention here a few. The authors say that there is little consensus on the contributions of these two components to the decrease in Q. I would say that there is plenty, mainly afforestation and/or flow regulation. And what is the Qh have to do with the bias correction. Again, please expand on this.

We think the methodology section is clearer through incorporating the changes listed above in response to your comment 1, particularly with the inclusion of the flow chart. However, we do feel that the introduction as it is contains only information that is crucial to introduce the overarching ideas and, namely, the Budyko framework. While used in the methodology, the water balance (equation 1) is introduced here because it is critical in discussing the partitioning of $P$ into $Q$ and $E$ and how the Budyko formula helps to understand this. While we refer to equations 2-3 in the methodology to derive further equations, they are not directly used in any calculations in the form presented in the introduction. The data section only contains information on the data used, with equation 4 simply stating that we calculate $E_p$ in the CMIP5 models directly from net surface radiation (CMIP5 data that is archived). Also the methods section is currently in chronological order in the sense that one needs to determine $\omega$ (equation 5), before calculating $Q_a$ (equation 6), before $Q_a$ can be used in the decomposition of $Q_m$ in equation 7 (with $Q_h$ also detailed in the section above equation 7). In addition to the improvements to the structure of the methodology suggested above, we have added some specific additional information to make the manuscript easier to follow. For example, the following sentence has been added to the paragraph underneath equation 8:

"In Eq. (8) we use $\omega$ values calculated using observed data and Eq. (5) for the 1951–2000 period (1.77 and 2.44 for the Yangtze and Yellow, respectively)."

MANUSCRIPT AMENDED.

And we have moved the paragraph talking of comparing directly simulated $Q$ with $Q$ calculated from $P - E$ from the data section 2.1.2 to the methods section 2.2.2, because it fits better under methods. We have also rewritten it to be clearer:

"We compare $Q^*$ with the original CMIP5 model simulated $Q$, calculated as $P - E$. Data for $Q$ are also directly available for 28 of the 34 GCMs. Conclusions should not be sensitive to using either direct $Q$ output or water balance-derived $Q$ if changes in storage are negligible. Bring et al. (2015), however, showed evidence for long-term systematic changes in water storage in some CMIP5 models. Although not a primary analysis, it is sensible to test the sensitivity of our results to the choice of $Q$."

MANUSCRIPT AMENDED.

Of the Budyko/Chinese basins studies that you list, only one is directly related to the Budyko framework. However, we agree that we should mention this study and some other high profile papers that we found. To the page 3, line 10 paragraph we have added:

"There is a wealth of literature that uses the Budyko framework to understand water changes in other Chinese basins (Yang et al., 2007; Xu et al., 2014; Liang et al., 2015)."

b-c; Yes, for the historical 20th century analysis, which is more clearly separated in the revised manuscript.

d; We agree that this could be easier to understand. We have therefore added some clarification. You are correct in saying it is meant to represent climatic $Q$, or rather changes due to aridity change alone. It does also include a representation of some non-climatic factors but we expect these to be of second-order importance compared to changes in $P$ and $E_p$. Therefore, it should agree with the calculation of $Q$ using the Budyko framework, which also accounts for aridity change alone. To the page 4, line 15 paragraph in the introduction we have added:

"If these estimates reconcile it suggests that the Budyko framework is suitable for this attribution, since $Q$ simulated by the LSM should largely reflect changes in $P$ and $E_p$ only."

MANUSCRIPT AMENDED.

In the 20th-century historical changes data section (section 2.1.1) we have highlighted that we consider a second LPJ LSM run that does not include land-use changes and we can therefore test the assumption that this specific non-climatic factor has a negligible effect on simulated $Q$ compared to $P$ and $E_p$. To the page 6, line 30 paragraph we have added clarification of how this extra run provides an extra line of evidence but does not form a main focus of the paper:

"The run used in our primary analyses was also driven by historical land-use changes, calculated from the History Database of the Global Environment (HYDE) (Klein Goldewijk and Verburg, 2013). A separate run excludes the HYDE dataset, so that we are able to test the sensitivity to land-use changes. Assuming that any sensitivity is minimal, we only comment on this separate run briefly."

MANUSCRIPT AMENDED.

Further, in the 20th-century historical changes methods section (section 2.2.1) we define the runoff simulated by the LPJ LSM as $\Delta Q_{a_l}$ and added extra information to

explain what this represents and how land-use changes are briefly considered:

"That is to say, $\Delta Q_{a_l}$ should be dominated by changes in $P$ and $E_p$ and show strong agreement with $\Delta Q_a$. We specifically test the sensitivity to land-use changes since they are excluded in a separate run of the LPJ LSM model. This is the only change between the two runs, so we can elucidate the influence of land-use changes by simply taking the difference between them."

MANUSCRIPT AMENDED.

e-f; Yes, although we actually use 1951–2100 (page 9, line 5 paragraph).

g; This is explained in the page 7, line 20 paragraph and we actually use equation 5.

h-i; Yes. The calculation of $Q_h$ is explained at the bottom of page 6/top of page 7. It has nothing to do with the bias correction, which concerns the projected CMIP5 analysis. We hope this is more clear with the inclusion of the flow chart and improved methodology section as described above.

j-k; Yes. The LJP LSM runoff should be a proxy for $Q_a$ calculated via the Budyko framework as detailed in the text (page 8, line 30 and supplementary Sect. S3).

l; Again, this is concerning the projected 21st century CMIP5 analysis, which is much clearer in the revised manuscript.

The authors need to mention what calculations are related with the bias-correction, and which ones are related with the aim of calculating the human component of Q (Qh). So, is this the chronological order of calculations? If not, please modify. Also, where is the correction bias coming into these methods, I could not see it, until maybe the calculation of Q*. Please specify.

We hope the revised layout of the methodology makes this much clearer. We have included the equations that are used in each calculation and in each application in Fig. 1.

Response to anonymous referee #2

**Thank you for reviewing our paper and providing suggestions that have improved it. Our responses to your comments are in bold font. Where we reference pages, lines, figure and tables, this corresponds to the revised maunscipt and approximately to the tracked changes manuscript.**

The manuscript is generally in a good shape, well structured and well written. The overall presentation of the results is good with concise and high-quality figures. The methodological approach seems to be technically sound, but (due to its complexity) needs to be explained better. Maybe some sort of conceptual figure or flow chart would help! Further, the obtained results depend on many assumptions which potentially not permit a robust interpretation of the results. The authors already discuss some limitations of their approach, but it is my assessment that this discussion needs to be extended before final publication. Also, some important references are missing to better outline some issues and limitations of their approach.

**The other referee has also commented that the methodological approach could be explained better. We have therefore added a flow chart (Figure 1) to the end of the introduction section in the revised manuscript to serve as an overview for the subsequent breakdown of the paper.**

**The key message from this is that the Budyko curve, or rather the updated Fu version (equation 3), can be used to both attribute past changes and help refine future changes. These two strands share common ideology but are attempting to tackle different problems and therefore use different equations. The flow chart more clearly emphasises this and even lists the equations that are used in each application. We have also divided the Data (section 2.1) and Methods (section 2.2) subsections into further subsections to more clearly differentiate the two applications.**

**MANUSCRIPT AMENDED.**

**Major comments:**

1) I would be very careful with separating the measured change in runoff into the individual components as done in eq. 7. If you assume such a linear relationship, you also assume the individual components to be independent, which they are clearly not! Especially the separation

**Introducing ideas**
**Section 1**

The partitioning of precipitation (Eq. (1)) and the non-parametric and one-parameter versions of the Budyko curve (Eqs. (2) and (3), respectively).

[Figure]

**Applying the
Budyko framework**

**20th century historical changes**
**Sections 2.1.1, 2.2.1 and 3.1**

1) Use Eq. (5) to calibrate ω, using observed $P$, $E_p$ and $E$ ($E$ is calculated using observed $P$ and $Q$ (Eq. (1)).

2) Calculate $Q_a$ using Eq. (6).

3) Estimate $Q_h$ using time series of water consumption derived from time series of Chinese irrigated area.

4) Separate the measured runoff changes into $Q_a$, $Q_h$ and a residual term using Eq. (7).

5) Use $Q$ as simulated by a LSM, to test the calculation of $Q_a$.

**21st century projected changes**
**Sections 2.1.2, 2.2.2 and 3.2**

1) Estimate $E_p$ in CMIP5 models from net surface radiation (Eq. (4)) .

2) Use Eq. (5) to calibrate ω, using observed $P$, $E_p$ and $E$ ($E$ is calculated using observed $P$ and $Q$ (Eq. (1)).

3) Bias correct $P$ and $E_p$ using Eq. (10).

4) Use bias corrected $P$ and $E_p$, together with the calculated ω values to calculate $Q^*$, a Budyko corrected runoff. This uses Eq. (8).

**Agreement of 20th century changes with existing literature will validate the use of the Budyko framework in refining 21st-century projections**

Figure 1: Schematic of how the Budyko framework is used to improve our understanding of 20th-century historical changes and 21st-century projected changes.

into Qh and Qo is potentially dangerous. Please also be aware that in the context of the Budyko framework, aridity is solely defined through the notional, dimensionless ratio Ep/P, which has no direct physical meaning. Everything else besides mean annual Ep/P is actually integrated into w. Also, w and Ep/P are not necessarily independent (please see Padron et al, 2017). It would be nice if you could try to determine if there are dependencies between Qa, Qh, and Qo. Is it possible to plot these against each other? In case there are large dependencies and interrelationships the obtained results might be less meaningful.

**This is a very good point and something that we have explored further. Unfortunately it is difficult to test for dependencies between $Q_h$ and $Q_o$ due to the limited temporal resolution (decadal mean values) of the irrigated area time series of Freydank and Siebert (2008) that is used to calculate $Q_h$ (page 13, line 10). We can, however, look at the relationship between $Q_a$ and changes due to all other factors besides aridity change ($\omega$), represented by the residual $\Delta Q_h + \Delta Q_o$ in equation 7. The interannual correlation is -0.35 (Figure 2).**

[Figure]

Figure 2: Interannual correlation.

**However, this correlation becomes positive (0.66) when considering 5-year means (Figure 3). Although both correlations are significant, the change in sign leads us to believe that there is not strong and consistent evidence for such a relationship here. We do feel it is important to point this out as a potential limitation of decomposing changes in runoff into these separate components. We have therefore added the**

[Figure]

Figure 3: 5-year mean correlation.

following to the discussion (page 21, line 30):

"It is also important to note some potential limitations of using Eq. (7) to separate the measured decrease in Yellow river runoff into various components. This approach assumes a linear relationship and therefore that the individual components are independent. Padron et al. (2017) showed that cross correlations exist between many of the factors suggested to influence runoff through $\omega$. Testing for dependencies between $\Delta Q_h$ and other components is unfortunately limited by the poor temporal resolution of the irrigated area time series of Freydank and Siebert (2008). Although we find that interannual variations in $Q_a$ and the residual $Q_h + Q_o$ are correlated (-0.35), this correlation is weak and reverses sign when considering multi-year means. Further, our approach considers long-term trends/changes in runoff, which means that any dependencies at shorter timescales should not influence conclusions."

MANUSCRIPT AMENDED.

2) In the main text, the authors assume w (omega) to be constant. In the Supplementary they further present results obtained for a (time-)varying w. I actually leave this to the authors, but I would almost prefer to present the time-varying more prominently. A certain variation around the original Budyko curve and thus a variation of w is actually inherent to the Budyko framework. This was already stated by Budyko himself. Hence, the Budyko framework is not necessarily

deterministic. There is a growing body of literature interpreting the Budyko framework in a probabilistic sense (e.g. Greve et al., 2015, Singh and Kumar, 2015, Gudmundsson et al., 2016, etc.), thus accounting for the spread in w and taking into account the complex interplay of all other factors (besides the aridity index). By using the time-varying approach you basically account for these variations, which in my assessment is more realistic.

**We were equally unsure during the writing of the paper whether to present the time-varying analysis in the manuscript or leave it as supplementary material. Although, as you say, using the time-varying approach allows a more complete assessment of changes within the Budyko framework, our main aim here is to illustrate the large improvements that can be made by considering CMIP5 output without the large aridity biases that are currently present. The choice of $\omega$ (constant or time-varying) actually plays a small role in shaping changes, compared to correcting for aridity biases ($E_p/P$). Also, many of the other factors that determine $\omega$ are not well represented in CMIP5 models so that a full consideration of $\omega$ is difficult. We understand the argument for both choices but, for the reasons given above, have decided to keep the time-varying approach in the supplementary materials.**

3) In this context, please be aware and discuss that you are considering temporal variations here. Most of the referenced Budyko-based studies actually consider spatial variations. It is important to note that these are not necessarily tradable (Berghuijs and Woods, 2017).

**This is an important point to make. We would like to point out that the year-to-year variability calculated through the Budyko framework and presented in Figs. 5 and 10 should not be taken at face value, since changes in storage could dominate at these timescales. The Budyko framework is more appropriately applied to study long-term mean changes, which is exactly what we do with the CMIP5 projected changes. Although we present year-to-year variability in Fig. 10, the conclusions are drawn from the differences between two 20-year means at the end of the 20th and 21st centuries. Likewise, we look at trends in the 20th century analysis. It is worth pointing out that the conclusions on historical changes, regarding the contributions of aridity change and direct human impacts, are qualitatively the same if considering differences between 10 (or 20) year means at the start and end of the 1951–2000 period. We have pointed this out in the text by adding the following to the end of**

the discussion (page 22, line 25):

"Most applications of the Budyko framework consider spatial rather than temporal variations. Berghuijs and Woods (2016) demonstrate that spatial and temporal variations are not necessarily tradable. We stress that the Budyko framework is not employed here to robustly determine interannual variability in water availability, but is instead used to understand long-term trends (Sect. 3.1) or the difference between 20-year means at the end of the 20th and 21st centuries (Sect. 3.2)."

And the following to the text underneath equation 7 (page 8, line 20):

"...with changes over the historical period (1951–2000) calculated as the linear trend. We note that our conclusions are not affected by using the difference between either 10 or 20-year means at the beginning and end of the historical period."

MANUSCRIPT AMENDED.

4) You use the Budyko-based equation 8 to compute Q* for each year. One of the main assumptions of the Budyko hypothesis is stationarity, i.e. storage changes are negligible. This might, however, not be the case here due to interannual variations in storage. Have you maybe tested to use longer time periods to smooth out long-term variations in water storage? How would your results look like when computing decadal Q*?

Because we have no information on changes in storage when we calculate $Q$ as $P - E$ (and assume changes in storage are equal to zero) our results do not change through taking decadal means (Figure 4).

We actually check the potential impact of storage changes by considering values for a subset of 28 (from 34) CMIP5 models for which $Q$ is directly simulated. This information is shown in Fig. 10 and Table 1 and does not affect conclusions, especially for the Yellow catchment, as pointed out in the results.

Minor comments:

These are all useful comments and we have acted on all of them.

p. 1, l. 3: I would not use "describe" here, it might be better to use something like "represent". The Budyko framework does not mechanistically describe the relationship between the aridity

[Figure]

Figure 4: Using decadal means.

and the evaporative index.

**Changed to "represent".**

**MANUSCRIPT AMENDED.**

p. 1, l. 16-19: There is some recent literature which challenges the referenced papers and the ubiquitous increase in aridity in a warmer world (Roderick et al., 2015, Scheff et al., 2017, Greve et al., 2017, Scheff, 2018). Might be worth to mention that this is an ongoing debate.

**Changed to:**

**"At the largest scales, the majority of literature on projected changes in aridity suggests a global land drying tendency (Dai, 2013; Cook et al., 2014; Scheff and Frierson, 2015); a consequence of ubiquitous increases in potential evapotranspiration (Ep), but mixed signals in precipitation (P ). This has been challenged, however, by some recent studies (Roderick et al., 2015; Greve et al., 2017; Scheff et al., 2017)."**

**MANUSCRIPT AMENDED.**

p. 2, l. 1-3: It might be better to rephrase this a bit. Depending on the context, the terms "water supply" and "water demand" are interpreted differently. Water supply is not necessarily just atmospheric water supply. Water supply can also be runoff. Some people also consider groundwater or water from other, unconventional sources (water transfer, desalination, etc.) as water "supply". Also, water stored in reservoirs is some sort of water supply. Water demand is often used in terms of human water consumption, including domestic and industrial water use as well as water used for irrigation.

**We agree that this was poor wording. Rather than considering the entire problem in this context we rather meant that most studies working on water availability projections consider very specific components of supply and demand. We have rewritten the opening sentence as this (page 2, line 5):**

**"Literature on future water availability projections has typically been framed around the net atmospheric supply of water versus the net demand for water resulting from direct human impacts (land-use change, dam construction and reservoir operation, and surface water and groundwater consumption for irrigation)..."**

**MANUSCRIPT AMENDED.**

p. 3 l. 7: For w=2.6, please cite Yang et al, 2008.

**MANUSCRIPT AMENDED.**

p. 4, Fig. 2: The basin outlines look a bit strange. I know that this is based on grid cells, but why dont you just outline the grid cells instead of interpolating the basin borders. This would also help to avoid the gap between both basins

**We have changed the plots to avoid this gap between the Yellow and Yangtze basins.**

**MANUSCRIPT AMENDED.**

p. 4, l. 3: Among seasonality, snow dynamics and storminess it is also many other factors that are related to a hydroclimatological change. Maybe just add an "etc."

**Changed to:**

**"The contributions of climate change (which incorporates aridity change, but also changes in seasonality, snow dynamics, storminess and many other factors..."**

**MANUSCRIPT AMENDED.**

p. 5, l. 24-26: Is this shown by Sitch et al., 2013. In this case, it might be better to add the reference at the end of the sentence. Or is this based on a separate analysis? In this case, you might add something like "not shown"?

**Good point. Sitch et al. (2013) referred to the TRENDY intercomparison project. The runoff coefficient comment was based on our own separate analysis, so we have added a "not shown" comment.**

**MANUSCRIPT AMENDED.**

p. 6, eq. 4: Milly actually proposes an ad-hoc multiplier of 0.8. Have you checked if you can maybe reduce some of the biases in Ep by adjusting the multiplier?

**This does affect $E_p$ biases. The multi-model mean bias for $E_p$ for the Yellow catchment changes from +0.30 mm/day to -0.25 mm/day relative to the observed climatology. But since the aridity biases are largely a result of precipitation biases, this makes little**

**difference to the overall results ("Figure" 5; compare to Table 1 of the manuscript).**

|  |  | RCP4.5 | RCP8.5 |
|---|---|---|---|
| Yangtze (all): | $\Delta Q$ | 0.12 +/- 0.32 | 0.14 +/- 0.40 |
|  | $\Delta Q^*$ | 0.18 +/- 0.34 | 0.20 +/- 0.41 |
| Yangtze (subset): | $\Delta Q$ | 0.08 +/- 0.26 | 0.09 +/- 0.35 |
|  | $\Delta Q^*$ | 0.13 +/- 0.27 | 0.15 +/- 0.34 |
|  | $\Delta Q_{direct}$ | 0.08 +/- 0.25 | 0.10 +/- 0.33 |
| Yellow (all): | $\Delta Q$ | 0.07 +/- 0.11 | 0.09 +/- 0.14 |
|  | $\Delta Q^*$ | 0.07 +/- 0.07 | 0.10 +/- 0.10 |
| Yellow (subset): | $\Delta Q$ | 0.06 +/- 0.11 | 0.09 +/- 0.15 |
|  | $\Delta Q^*$ | 0.06 +/- 0.08 | 0.10 +/- 0.10 |
|  | $\Delta Q_{direct}$ | 0.06 +/- 0.11 | 0.09 +/- 0.16 |

Figure 5: The equivalent to Table 1 in the manuscript using the Milly and Dunne (2016) multiplier.

p. 8, l. 30: See my first major comment. This might also be due to dependencies between each factor.

**We have covered this in the discussion.**

p. 18, Fig. 10: Is the time series for Q* computed annually? And is the 5yr running mean subsequently computed from the annual time series? Or is Q* computed from the 5yr running mean time series of P and Ep?

**Yes, it is computed annually and the running mean is computed from the annual time series.**

p. 19, l. 30-32: Please note here that Li et al. (2018) is potentially only valid at large catchment scales. At other scales, this might not be the case. You might also note here or somewhere else that the interpretation of w is still largely and inconclusive and sometimes also contradictory (Padron et al., 2017).

**We have added the important caveat that this vegetation link may only hold for large catchments..**

**MANUSCRIPT AMENDED.**

[revised manuscript text omitted]